# Concept-Based Abductive and Contrastive Explanations for Behaviors of Vision Models

## Abstract

*Concept-based explanations* offer a promising approach for explaining the predictions of deep neural networks in terms of high-level, human-understandable concepts. However, existing methods either do not establish a causal connection between the concepts and model predictions or are limited in expressivity and only able to infer causal explanations involving single concepts. At the same time, the parallel line of work on *formal abductive and contrastive explanations* computes the minimal set of input features causally relevant for model outcomes but only considers low-level features such as pixels. Merging these two threads, in this work, we propose the notion of *concept-based abductive and contrastive explanations* that capture the minimal sets of high-level concepts causally relevant for model outcomes. We then present a family of algorithms that enumerate all minimal explanations while using *concept erasure* procedures to establish causal relationships. By appropriately aggregating such explanations, we are not only able to understand model predictions on individual images but also on collections of images where the model exhibits a user-specified, common *behavior*. We evaluate our approach on multiple models, datasets, and behaviors, and demonstrate its effectiveness in computing helpful, user-friendly explanations.

## 1 Introduction

Concept-based explanations (ConXps) for deep neural networks used in vision tasks relate the outputs (*behaviors*) of models with internal neural representations that encode human-understandable semantic concepts or symbols. ConXps are computed in two steps—first, the internal encoding of concepts are inferred, typically, as vectors in the activation space of the models; second, the impact of the concepts on the model outputs is determined. The type of impact analysis dictates the nature of the ConXps. A seminal approach Kim et al. (2018) estimates the sensitivity of a model's output to changes in concept activation values[1]. A limitation of such approaches is that they do not allow one to judge if the activation of a concept is necessary or sufficient for the model's output. Other approaches use *concept erasure* methods Bhalla et al. (2024); Belrose et al. (2023); Holstege et al. (2025); Ravfogel et al. (2022) to determine the necessity of a concept for the model's output by erasing the concept from the internal activations on an input. If erasing the concept changes the output, then the concept is necessary. However, these methods only consider the impact of individual concepts and are not able to judge when multiple concepts are necessary for changing (or sufficient for retaining) the model output.

A parallel line of work on *formal abductive and contrastive explanations* Ignatiev et al. (2020b); Marques-Silva & Ignatiev (2022); Bassan et al. (2026) computes the minimal sets of sufficient (abductive) and necessary (contrastive) features for a model's output. Each explanation can comprise of more than one feature. These explanations are *formal* since they guarantee that the inferred sets of features are indeed sufficient/necessary and minimal. Computing such explanations is typically reduced to solving constraint satisfaction problems via tools such as SAT and SMT solvers. However, these methods have been applied only in the context of low-level features such as pixels, which often lack conceptual meaning. These methods also struggle to scale to state-of-the-art models because of the costs of constraint solving.

---

[1] *Concept activation value* (or *strength*) is the degree to which a concept is activated for a particular input. See Definition 2.5

To address the limitations of these previous lines of work, we propose the notion of *concept-based abductive and contrastive explanations*. Concept-based abductive explanations (ConAXps) are minimal sets of concepts such that retaining these while erasing all other concepts from the representation keeps the model behavior unchanged. Concept-based contrastive explanations (ConCXps) are minimal sets of concepts such that erasing these while retaining all other concepts changes the behavior.

To infer internal representations of concepts, we leverage known techniques Mangal et al. (2024); Dreyer et al. (2025) that use vision-language models, such as CLIP Radford et al. (2021), as a semantic lens for understanding the representations learned by a (separate) vision model. These approaches rely on three key observations. First, extracting concept vectors from text encoders is cheap and requires no labeled data—one can simply perform a forward pass through the text encoders using sentences that describe the relevant concept to yield a concept vector. Second, CLIP's representation space is shared by its text and image encoders such that embeddings of images and respective text captions are aligned. Third, the internal representation spaces of vision models and CLIP can be accurately mapped to each other.

Given concept vectors, we present three different algorithms to compute ConXps: (1) a naive algorithm (NaiveEnum) that exhaustively enumerates every possible set of concepts, starting from sets with single concepts, and checks if the set is a ConAXp or ConCXp; (2) an algorithm (XpEnum) adapted from the formal explanations literature Ignatiev et al. (2020b) that uses the notion of hitting sets and exploits a known duality between abductive and contrastive explanations; and (3) a novel algorithm (XpSatEnum) that is explicitly geared towards finding explanations that are common across a given set of images. All three algorithms are parametric with respect to the concept erasure procedure. In this work, we evaluate three different concept erasure methods. The methods differ in how they erase the concept from the representation. Two of the methods have been proposed in prior work Bhalla et al. (2024); Belrose et al. (2023). The new erasure method we propose is based on the intuition that after erasure, the representation should be orthogonal to all the concept vectors that represent the erased concepts while being as similar as possible to the original representation.

We evaluate our approach on three different vision classifiers, namely, zero-shot CLIP, ResNet18, and VGG19 and two datasets (RIVAL-10 Moayeri et al. (2022) and EuroSAT Helber et al. (2019)). To obtain explanations that are *general*, i.e., they explain model outputs on sets of images where the model exhibits a common behavior, we aggregate the ConXps computed per image to construct a histogram of ConXps as the explanation for model behavior on a set of input images. Figure 1 gives an example of the types of explanations we are able to infer. Our experiments reveal that the ConXps computed using our methods are sparse—a few short explanations are able to explain model output for a large majority of the images in the considered set. Our explanations are generalizable in the sense that they are applicable even for unseen images where the model exhibits the same behavior. Moreover, the concept-based nature of the explanations makes it easy to judge if the model relies on relevant and irrelevant concepts.

## 2 Preliminaries

The paper assumes a deep neural network $M^{vis}$ trained to solve a visual classification task. The techniques we propose, however, are generic and applicable to models designed for other tasks. Model inputs are images from a set $Img$. The model output is a prediction $k \in \mathcal{K}$ where $\mathcal{K} = \{k_1, k_2, \ldots, k_m\}$ is a set of classes. Model $M^{vis}$ is a function of type $Img \rightarrow \mathcal{K}$. Throughout the paper, we will assume that indices start from 1.

### 2.1 Formal Explanations

For the purpose of this subsection, we assume that the input of the model consists of a set of features $\mathcal{F} = \{f_1, f_2, \ldots, f_n\}$, each feature $f_i$ taking values from a domain $D_i$. Thus, our feature space is defined by $\mathbb{F} = \prod_i D_i$. Model $M$ is a function of type $\mathbb{F} \rightarrow \mathcal{K}$. An instance $(v, k)$ is an assignment of values to features, $v = (v_1, v_2, \ldots, v_n) \in \mathbb{F}$, that leads to a prediction $k = M(v), k \in \mathcal{K}$.

**Definition 2.1** (Weak Abductive Explanation)**.** Given an instance $(v, k)$ s.t. $M(v) = k$, a weak abductive explanation (WeakAXp) is a subset of features $\mathcal{X} \subseteq \mathcal{F}$ which, if unchanged, is sufficient to ensure prediction

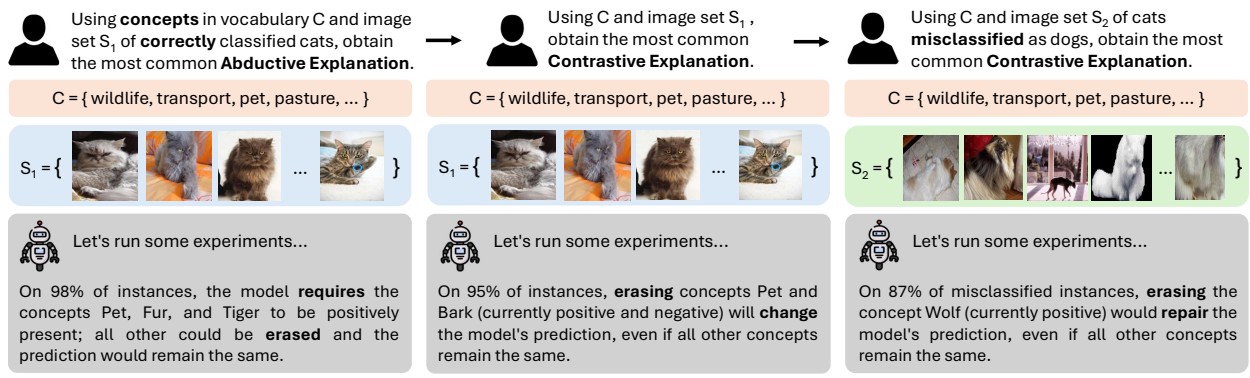

Figure 1: Examples of concept-based explanations of model behaviors. Concept polarity is determined by the sign of the concept activation values. **Left:** Abductive explanations identify the minimal required concepts for correct classification in set $S_1$. **Middle:** Contrastive explanations highlight concepts whose erasure triggers a prediction change. **Right:** Analysis on misclassified set $S_2$ identifies specific concept-based repairs to rectify the model's decision.

$k$, irrespective of the value taken by the features not in set $\mathcal{X}$. We can think of WeakAXps as answers to the question: *Why is the prediction $k$?* Ignatiev et al. (2020b).

$$\texttt{WeakAXp}(\mathcal{X}, M, v, k) := \forall (x \in \mathbb{F}). \bigwedge_{j \in \mathcal{X}} (x_j = v_j) \rightarrow (M(x) = k)$$

**Definition 2.2** (Abductive Explanation). Given an instance $(v, k)$ s.t. $M(v) = k$, an abductive explanation (AXp) is a subset-minimal set of features $\mathcal{X} \subseteq \mathcal{F}$ which, if unchanged, is sufficient to ensure prediction $k$. In other words, an AXp is a subset-minimal WeakAXp, meaning that it doesn't contain a smaller subset that can itself be a WeakAXp.

$$\texttt{AXp}(\mathcal{X}, M, v, k) := \texttt{WeakAXp}(\mathcal{X}, M, v, k) \wedge (\forall (\mathcal{X}' \subsetneq \mathcal{X}). \neg \texttt{WeakAXp}(\mathcal{X}', M, v, k))$$

Intuitively, we can think of an AXp as a WeakAXp that can not be further shrunk; no element can be further subtracted from it without failing to ensure that the original prediction holds.

**Definition 2.3** (Weak Contrastive Explanation). Given an instance $(v, k)$ s.t. $M(v) = k$, a weak contrastive explanation (WeakCXp) is a subset of features $\mathcal{X} \subseteq \mathcal{F}$ which, if changed, is sufficient to change the prediction $k$. We can think of WeakCXps as answers to the question: *'How can the prediction be changed to $\neg k$?'* Ignatiev et al. (2020b).

$$\texttt{WeakCXp}(\mathcal{X}, M, v, k) := \exists (x \in \mathbb{F}). \bigwedge_{j \notin \mathcal{X}} (x_j = v_j) \ \wedge \ (M(x) \neq k)$$

**Definition 2.4** (Contrastive Explanation). Given an instance $(v, k)$ s.t. $M(v) = k$, a contrastive explanation (CXp) is a subset-minimal set of features $\mathcal{X} \subseteq \mathcal{F}$ which, if changed, is sufficient to change the prediction $k$. Analogous to AXps, a CXp is a subset-minimal WeakCXp, meaning that it doesn't contain a smaller subset that can itself be a WeakCXp.

$$\texttt{CXp}(\mathcal{X}, M, v, k) := \texttt{WeakCXp}(\mathcal{X}, M, v, k) \wedge (\forall (\mathcal{X}' \subsetneq \mathcal{X}). \neg \texttt{WeakCXp}(\mathcal{X}'))$$

In the same sense, we can think of a CXp as a WeakCXp that can not be further shrunk; no element in the CXp can be allowed to retain its original value without failing to change the original prediction.

## 2.2 Internal Representations of Concepts

Our goal is to reason about model behaviors via explanations containing sets of human-understandable concepts. For this, we leverage the *linear representation hypothesis* Kim et al. (2018); Park et al. (2024);

Elhage et al. (2022), i.e., the observation that DNNs learn to represent semantic concepts as directions in their representation spaces These concept representations are referred to as *concept vectors*. Formally, given a model $M^{vis}$, the hypothesis states that the model can be decomposed as $M^{vis} = M^{head} \circ M^{enc}$ where $M^{enc} : Img \rightarrow \mathbb{Z}$, $M^{head} : \mathbb{Z} \rightarrow \mathcal{K}$, and $\mathbb{Z}$ is the representation (or embedding) space. The encoder translates low-level features (e.g., pixels in an image) into high-level representations and the head predicts an appropriate class $k \in \mathcal{K}$ based on these representations. Concept representations are vectors in the space $\mathbb{Z}$. Given any model embedding $z \in \mathbb{Z}$, we can compute the *strength* of each concept in $z$. A number of methods Kim et al. (2018); Mangal et al. (2024) have been proposed in the literature for, both, extracting concept vectors from vision models and computing the *strength* of concepts in image embeddings.

### 2.2.1 CLIP-based Representation Surrogates

In this work, we leverage methods that use the vision-language model CLIP as a semantic lens to understand the representations learned by vision models. These methods rely on the observation that there exist linear mappings between the embeddings of arbitrary vision models and the joint vision-language embeddings of CLIP Moayeri et al. (2023). The multi-modal nature of the CLIP embedding space makes it straightforward to extract concept vectors in the CLIP space without needing concept-labeled data. The linear maps between the embedding spaces allow analyzing the semantic content of vision model embeddings by mapping them to the CLIP space.

**Extracting Concept Vectors in CLIP Space.** As in past work Mangal et al. (2024), we use the CLIP text encoder to extract concept vectors. For a given concept, we construct a set of captions referring to that concept (see Appendix D for an example). These captions are all passed through CLIP's text encoder and the resulting embeddings are averaged and normalized to obtain the corresponding concept vector. As observed by Liang et al. (2022), there exists a modality gap between the image and text embeddings in CLIP's joint embedding space—these embeddings lie in separate cones. To address this gap, following Bhalla et al. (2024), we *mean-center* the image and the concept embeddings (see Appendix D for details).

**Mapping from Vision Models to CLIP and Back.** We frame the problem of learning the map $Wz+d$ from the vision model to the CLIP embedding space as the following optimization problem:

$$W, d := \underset{W,d}{\arg\min} \frac{1}{|D_{train}|} \sum_{x \in D_{train}} \|W \cdot M^{enc}(x) + d - M^{CLIP}_{img}(x)\|_2^2. \tag{1}$$

where $D_{train}$ is the set of training images and $M^{CLIP}_{img}$ is CLIP's image encoder. The map from CLIP space to vision model space is learned similarly. We design a sanity check to judge the quality of these maps. See Appendix D.1 for details.

### 2.2.2 Concept Strengths and Erasure

Intuitively, the notion of concept strength or concept activation value is the degree to which a concept is activated for a particular input. The concept strengths for all the concepts in the vocabulary together constitute a strength vector.

**Definition 2.5** (Concept Strength Vector). Given an image embedding $z \in \mathbb{Z}$, a vocabulary of concepts $\mathcal{C} = \{c_1, c_2, \ldots, c_n\}$ and their corresponding concept vectors $\vec{c_1}, \vec{c_2}, \ldots, \vec{c_n}$, the concept strength vector, $st(z)$, is given by $\forall i \in \{1, \ldots, n\}. \ st(z)_i := cos(z, \vec{c_i}).$[2]

Note that the *strength* of concept in the strength vector can be positive, negative, or zero.

Concept erasure ensures that a concept is no longer present in the embedding of an image.

**Definition 2.6** (Concept Erasure). Given an image embedding $z \in \mathbb{Z}$, a vocabulary of concepts $\mathcal{C} = \{c_1, c_2, \ldots, c_n\}$, and a concept $c_i$ to be erased, $erase(z, c_i) \in \mathbb{Z}$ is a $c_i$-erased embedding if $\forall j \in \{1, \ldots, n\}. \ st(erase(z, c_i))_j = 0$ if $i = j$ else $st(z)_j$.

---

[2]Concept strengths can be calculated using other methods such as via projection of the embedding onto the concept vectors or through a sparse linear decomposition of the embedding in terms of the concept vectors.

We will abuse notation and write $erase(z, b)$, where $b$ is a boolean vector of length $n$, to refer to an embedding $z$ where all the concepts $c_i$ such that $b_i = 0$ are erased. Note that, in practice, erasure procedures might also modify the strength of the concepts not being erased. We discuss this further in Section 4.1.

# 3  Concept-based Formal Explanations

We extend the existing notions of abductive and contrastive explanations to concept-based versions of such explanations. We focus on vision models that can be decomposed into an encoder and a head, i.e., $M^{vis} = M^{head} \circ M^{enc}$. For the purpose of these explanations, the input of $M^{head}$, which is an activation vector $z \in \mathbb{Z}$, is treated as being comprised of a set of concepts $\mathcal{C} = \{c_1, c_2, \dots, c_n\}$, each concept $c_i$ being present or not. Thus, the feature space of $M^{head}$ is defined by $\mathbb{B} = \prod_i \{0, 1\}$. An instance $(v, k)$ is an input image $v$ that leads to a prediction $k = M^{vis}(v), k \in \mathcal{K}$.

**Definition 3.1** (Weak Abductive Concept-based Explanation)**.** Given an instance $(v, k)$ s.t. $M^{vis}(v) = k$, a weak concept-based abductive explanation (WeakConAXp) is a subset of concepts $\mathcal{X} \subseteq \mathcal{C}$ which, if unchanged, is sufficient to ensure prediction $k$.

$$\texttt{WeakConAXp}(\mathcal{X}, M^{vis}, v, k) := \forall (b \in \mathbb{B}). \bigwedge_{j \in \mathcal{X}} (b_j = 1) \to (M^{head}(erase(M^{enc}(v), b)) = k)$$

**Definition 3.2** (Concept-based Abductive Explanation)**.** Similar to an abductive explanation (AXp), a concept-based abductive explanation is a subset-minimal set of concepts $\mathcal{X} \subseteq \mathcal{C}$ which, if unchanged, is sufficient to ensure prediction $k$.

$$\texttt{ConAXp}(\mathcal{X}, M^{vis}, v, k) := \texttt{WeakConAXp}(\mathcal{X}, M^{vis}, v, k) \land (\forall (\mathcal{X}' \subsetneq \mathcal{X}). \neg \texttt{WeakConAXp}(\mathcal{X}', M^{vis}, v, k))$$

**Definition 3.3** (Weak Contrastive Explanation)**.** Given an instance $(v, k)$ s.t. $M^{vis}(v) = k$, a weak concept-based contrastive explanation (WeakConCXp) is a subset of concepts $\mathcal{X} \subseteq \mathcal{C}$ which, if their presence or absence is changed, is sufficient to change the prediction $k$.

$$\texttt{WeakConCXp}(\mathcal{X}, M^{vis}, v, k) := \exists (b \in \mathbb{B}). \bigwedge_{j \notin \mathcal{X}} (b_j = 1) \ \land \ (M^{head}(erase(M^{enc}(v), b)) \neq k)$$

**Definition 3.4** (Concept-based Contrastive Explanation)**.** Similar to a contrastive explanation, a concept-based contrastive explanation (ConCXp) is a subset-minimal set of concepts $\mathcal{X} \subseteq \mathcal{C}$ which, if changed, is sufficient to change the prediction $k$.

$$\texttt{ConCXp}(\mathcal{X}, M^{vis}, v, k) := \texttt{WeakConCXp}(\mathcal{X}, M^{vis}, v, k) \land (\forall (\mathcal{X}' \subsetneq \mathcal{X}). \neg \texttt{WeakConCXp}(\mathcal{X}'))$$

The checks for `WeakConAXp` and `WeakConCXp` are exponential in the size of the concept vocabulary. For instance, ensuring `WeakConAXp` requires us to check for all possible combinations of boolean values for concepts not in set $\mathcal{X}$. We show below that, under a monotonicity assumption about the model head, these checks can be reduced to a single forward pass through the model. We provide empirical evidence that supports this assumption in Appendix A.1.

**Definition 3.5** (Monotonicity of Head)**.** Given a vision model $M^{vis}$, its head is monotonic with respect to concepts from a vocabulary $\mathcal{C} = \{c_1, c_2, \dots, c_n\}$ if adding a concept to a representation does not change the prediction of $M^{head}$ away from the original prediction $k = M^{vis}(v)$. Formally, for $k = M^{vis}(v)$,

$$\forall v \in Img, \forall b \in \mathbb{B}, \forall j \in \{1, \dots, n\}. \ M^{head}(erase(M^{enc}(v), b)) = k \to M^{head}(erase(M^{enc}(v), b[j \mapsto 1])) = k$$

where $b[j \mapsto 1]$ denotes the version of b where the $j^{th}$ index is 1 and the rest are unchanged. Intuitively, this indicates that concept $c_j$ should no longer be erased.

**Theorem 3.1** (Concept-based Explanations Under Monotonicity)**.** *Given a vision model $M^{vis}$, if we assume that $M^{head}$ is monotonic, then the definitions of weak concept-based abductive and contrastive explanations can be simplified as follows:*

$$\textit{WeakConAXp}(\mathcal{X}, M^{vis}, v, k) := \forall (b \in \mathbb{B}). \bigwedge_{j \in \mathcal{X}} (b_j = 1) \bigwedge_{j \notin \mathcal{X}} (b_j = 0) \to (M^{head}(erase(M^{enc}(v), b)) = k)$$

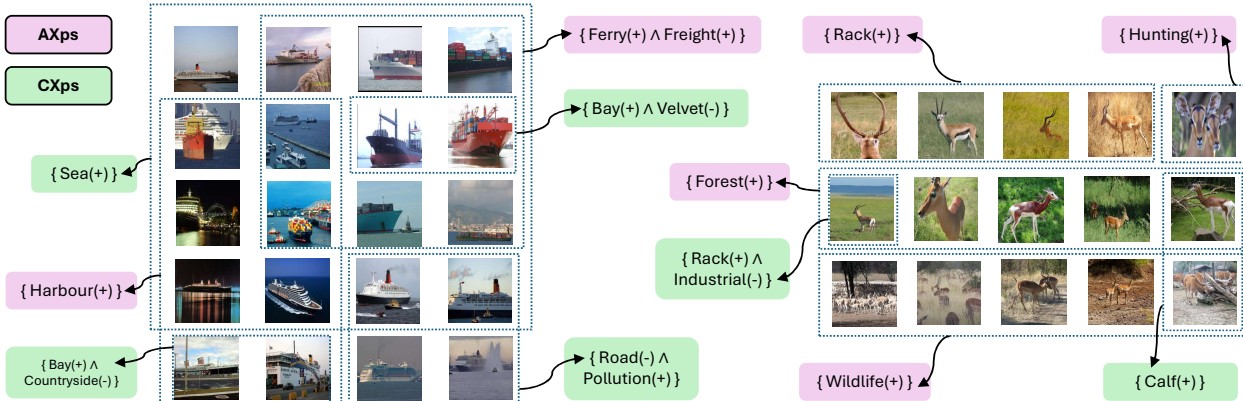

Figure 2: Concept-based abductive and contrastive explanations for correctly classified images of SHIP (left) and DEER (right). Abductive explanations (ConAXps, pink) identify concepts sufficient for the model's prediction, such as {Harbour(+)} or {Forest(+)}. Contrastive explanations (ConCXps, green) identify concept whose erasure would trigger a decision change, such as {Bay(+) ∧ Countryside(-)} or {Rack(+) ∧ Industrial(-)}. Different subsets of images within the same behavior are captured by distinct explanations.

$$WeakConCXp(\mathcal{X}, M^{vis}, v, k) := \exists(b \in \mathbb{B}). \bigwedge_{j \notin \mathcal{X}} (b_j = 1) \bigwedge_{j \in \mathcal{X}} (b_j = 0) \ \wedge \ (M^{head}(erase(M^{enc}(v), b)) \neq k)$$

*Proof.* See Appendix A. □

We define the notion of an *explanation enumerator* that enumerates all the ConAXps and ConCXps for a model $M^{vis}$ with respect to an image $v$. In the next section, we describe three different implementations of explanation enumerators that we use in this work.

**Definition 3.6** (Explanation Enumerator). An explanation enumerator $\mathcal{E}$ is a function that takes in a triple of model $M^{vis}$, input image $v$, and concept vocabulary $\mathcal{C}$ and returns a pair of sets of explanations, namely, all ConAXps and ConCXps for $M^{vis}$ with respect to $v$. Note that, for a single image $v$, there need not be a unique ConAXp and ConCXp.

In this work, rather than inferring concept-based explanations for individual instances $(v, k)$, our primary interest is in explaining *behaviors* of the model under analysis.

**Definition 3.7** (Behavior). Given a model $M^{vis} : Img \rightarrow \mathcal{K}$ and a dataset $Img$ where each instance has been labeled with one of a finite set of classes $\mathcal{K} = \{k_1, k_2, \ldots, k_m\}$, a behavior $B_{M^{vis}} \subseteq Img$ is a subset of images such that $\forall v_1, v_2 \in B_{M^{vis}}. f(v_1) = f(v_2)$ where $f$ is some function defined over the set $Img$.

In this work, we focus on two types of behaviors. First, *misclassification behavior* where the model $M^{vis}$ misclassifies images with label $k_i$ as label $k_j$— denoted as $B_{M^{vis}}(k_i, k_j)$ and formally defined as $f(v) := (M^*(v) = k_i) \wedge (M^{vis}(v) = k_j)$ where $M^*$ refers to the oracle labeling function. Second, *correct classification behavior* where $M^{vis}$ correctly classifies images with label $k_i$—denoted as $B_{M^{vis}}(k_i)$ and defined as $f(v) := (M^*(v) = k_i) \wedge (M^{vis}(v) = k_i)$.

**Definition 3.8** (Signed Explanation). Given a model $M^{vis}$, an input $v$, a concept vocabulary $\mathcal{C}$, and a concept-based explanation $\mathcal{X} \subseteq \mathcal{C}$, we define the signed version of $\mathcal{X}$ as the pair $(\mathcal{X}, smap)$ where $smap : \mathcal{X} \rightarrow \{+1, -1, 0\}$ is a map such that $\forall c_i \in \mathcal{X}. smap(c_i) = sgn(st(M^{enc}(v))_i)$. Here, $st$ is the concept strength vector for the embedding of $v$ and $sgn$ is the sign function.

Intuitively, the signed explanation is a set of concepts along with an indication of whether those concepts are positively or negatively present in the embedding. We will abuse notation and use $sgn(\mathcal{X}, M^{vis}, v)$ to refer to the *smap* of an unsigned explanation $\mathcal{X}$

Figure 2 shows examples of signed concept-based explanations. To explain a model behavior, we aggregate the set of signed explanations for each image in the behavior.

**Definition 3.9** (Concept-based Explanations of a Model Behavior)**.** Given model $M^{vis} : Img \rightarrow \mathcal{K}$, behavior $B_{M^{vis}}$, concept vocabulary $\mathcal{C}$, and explanation enumerator $\mathcal{E}$, the concept-based explanation of $B_{M^{vis}}$ is a pair $(h_A, h_C)$ such that

$$h_A((\mathcal{X}, smap)) = |\{v \in B_{M^{vis}} \mid \mathcal{X} \in \pi_1(\mathcal{E}(M^{vis}, v, \mathcal{C})) \ \wedge \ smap = sgn(\mathcal{X}, M^{vis}, v)\}|$$

$$h_C((\mathcal{X}, smap)) = |\{v \in B_{M^{vis}} \mid \mathcal{X} \in \pi_2(\mathcal{E}(M^{vis}, v, \mathcal{C})) \ \wedge \ smap = sgn(\mathcal{X}, M^{vis}, v)\}|$$

where $\pi_1$ and $\pi_2$ denote projections onto the first and second components of the pair returned by $\mathcal{E}$, i.e., the sets of ConAXps and ConCXps respectively. The map $h_A$ is a histogram over signed ConAXps and $h_C$ is a histogram over signed ConCXps: they aggregate the per-image explanations across the entire behavior, recording how frequently each explanation occurs while taking the signs into account..

## 4 Algorithms

Figure 3 gives an overview of our approach for computing ConXps. The approach relies on erasure and explanation enumeration algorithms which we describe in this section.

### 4.1 Erasure Algorithms

Various methods have been developed to erase concepts from model representations. All these methods attempt to strike a balance between ensuring successful erasure and minimizing the modifications to the original embedding. In this paper, we use and compare three different methods for concept erasure, namely, orthogonal erasure (Ortho), SPLiCE-style erasure (SPLiCE) Bhalla et al. (2024), and LEACE Belrose et al. (2023).

**Ortho.** We propose the Ortho erasure procedure guided by the intuition that the erased embedding should be orthogonal to all the concept vectors that represent the erased concepts while maintaining the scalar projections of the original embedding on the unerased concepts. This can be expressed as a constrained optimization with the following closed-form solution (see Appendix C for a detailed derivation),

$$r = z - C\,G^+(C^\top z - t),$$

where $r \in \mathbb{Z} = \mathbb{R}^d$ is the erased embedding, $z \in \mathbb{Z} = \mathbb{R}^d$ is the original embedding, $c_1, \ldots, c_n \in \mathbb{R}^d$ are *concept vectors* and $C = [\vec{c_1} \ \ \vec{c_2} \ \ \cdots \ \ \vec{c_n}] \in \mathbb{R}^{d \times n}$ is the matrix obtained by stacking them, $t \in \mathbb{R}^n$ is the target score vector where $t_j = 0$ if concept $c_j$ is to be erased otherwise $t_j = \vec{c_j} \cdot z$, $G = C^\top C$, and $G^+$ is the Moore-Penrose pseudoinverse of $G$.

**SPLiCE Bhalla et al. (2024).** This method first computes a sparse, nonnegative concept decomposition of the embedding $z \in \mathbb{Z} = \mathbb{R}^d$ by solving the following optimization problem,

$$w := \min_{w \in \mathbb{R}^n} \|w\|_0 \quad \text{s.t.} \quad \sigma(z) \cdot \sigma(w^\top C) > 1 - \epsilon$$

where, as before, $C \in \mathbb{R}^{d \times n}$ is the stack of concept vectors and $\sigma(x) = x/\|x\|_2$. Given the solution $w^*$ to the optimization problem, to erase a concept $c_j$, we first define $t \in \mathbb{R}^n$ as $t_i = w_i^*$ for $i \neq j$ and $t_j = 0$. Then, the erased embedding $r \in \mathbb{R}^d$ is given by $t^\top C$.

**LEACE Belrose et al. (2023).** This is a closed-form method which provably prevents all linear classifiers from detecting a concept after its erased from an embedding while changing the embedding as little as possible. The idea is to find an affine transformation to apply to embeddings $z$ such that, after application, there is zero covariance, as measured using the training data, between the erased embeddings and the binary random variable $\mathbf{c}_i$ representing the human-assigned present/absent labels for concept $c_i$ while ensuring that the original embedding is minimally modified as measured using a broad class of norms. This can be

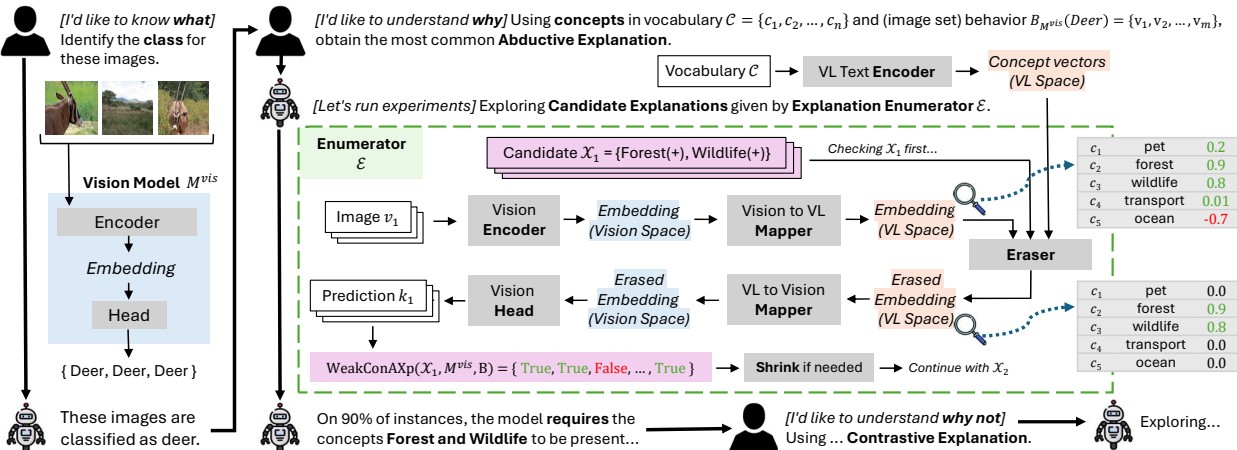

Figure 3: Overview of pipeline for finding concept-based explanations. The process begins with a vision model $M^{vis}$ exhibiting a behavior of interest, e.g. $B_{M^{vis}}(\text{Deer})$, correctly classified DEER images. To find abductive explanations comprising of concepts from vocabulary $\mathcal{C}$, an enumerator $\mathcal{E}$ explores a search space of size $2^{|\mathcal{C}|}$. For each candidate explanation $\mathcal{X}_i$, all images $v_i \in B$ are encoded and mapped into the embedding space of a Vision-Language (VL) model where vectors for each concept $c_i \in \mathcal{C}$ have previously been found. Then, an erasure algorithm is used to find a minimal modification of the embedding such that a subset of concepts, based on the current candidate explanation, are the only ones present (strength of all other concepts is reduced to zero). This modified embedding is mapped back to the vision space and passed through the vision model's head. A prediction change indicates that a weak abductive explanation has been found, which is shrunk if needed. The process for contrastive explanations is similar.

posed as the following constrained optimization problem, which Belrose et al. (2023) show has a data-specific closed-form solution.

$$P^* := \underset{P \in \mathbb{R}^d \times \mathbb{R}^d}{\arg\min} \|Pz - z\|_M^2 \quad \text{s.t.} \quad P \cdot Cov(\mathbf{z}, \mathbf{c}_i) = 0$$

where $Pz + b$ is the affine transformation applied to the original embedding $z$, $Cov(\cdot, \cdot)$ is the covariance matrix, and $\mathbf{z}$ is the random variable representing original embeddings.

## 4.2 Explanation Enumeration Algorithms

Given a model $M^{vis} = M^{head} \circ M^{enc}$, an instance $(v, k)$ where $v$ is the input image and $k = M^{vis}(v)$, and a vocabulary of concepts $\mathcal{C} = \{c_1, c_2, \ldots, c_n\}$, we present three different algorithms for computing the corresponding ConAXps and ConCXps. Given an instance $(v, k)$, the ConAXps and ConCXps for $M^{vis}$ are not unique. In fact, as we will later show in our experimental results, the number of explanations can be in the range of hundreds to thousands for realistic models. The space of all possible ConAXps and ConCXps is defined by $2^{\mathcal{C}}$. Each of the three presented algorithms takes a different approach to (bounded) exploration of this combinatorial space but they all assume that $M^{head}$ is monotonic which allows the use of the cheap checks defined in Theorem 3.1.

**NaiveEnum.** Algorithm 1 describes a naive approach that exhaustively enumerates all the elements of $2^{\mathcal{C}}$, starting from the empty set of concepts. While this method is guaranteed finding all ConAXps and ConCXps, it is computationally prohibitive for the large concept vocabularies with hundreds or thousands of concepts that are needed in practice. We use a depth-bounded search tree to maintain tractability. By fixing the maximum search depth at a value $K$, we limit the size of the resulting explanations to a maximum of $K$ concepts. Although this constraint excludes valid explanations whose size exceeds $K$, it is empirically justified (see Section 5.2.3) by the observation that the shortest explanations tend to be most effective, i.e., are most commonly shared across images on which the model under analysis exhibits the same *behavior* (for instance, all images of trucks that the model misclassifies as a car). To improve scalability, the algorithm

**Algorithm 1:** Depth-Bounded Naive Enum.

**Input:** Model $M^{vis}$, Image $v$, Vocabulary $\mathcal{C}$,
   Max. depth $K$
**Output:** Set of explanations $Xps$

1  $Xps \leftarrow \emptyset;\ cnds \leftarrow \{\emptyset\};$
2  **if** $\texttt{WeakConXp}(\emptyset, M^{vis}, v)$ **then**
3  $\quad$ **return** $\{\emptyset\};$
4  **for** $k = 1$ **to** $K$ **do**
5  $\quad cnds' \leftarrow \emptyset;$
6  $\quad$ **foreach** $cnd \in cnds$ **do**
7  $\quad\quad$ **foreach** $c \in \mathcal{C}$ **do**
8  $\quad\quad\quad cnd' \leftarrow cnd \cup \{c\};$
9  $\quad\quad\quad$ **if** $\texttt{WeakConXp}(cnd', M^{vis}, v)$ **then**
10 $\quad\quad\quad\quad Xps \leftarrow Xps \cup \{cnd'\};$
11 $\quad\quad\quad$ **else**
12 $\quad\quad\quad\quad cnds' \leftarrow cnds' \cup \{cnd'\};$
13 $\quad\quad cnds \leftarrow cnds';$

14 **return** $Xps;$

---

**Algorithm 2:** XpSatEnum

**Input:** Model $M^{vis}$, Behavior $B_{M^{vis}}$,
   Vocabulary $\mathcal{C}$, Initial explanations
   $\{Xps_v \mid v \in B_{M^{vis}}\}$
**Output:** Saturated explanations
   $\{Xps'_v \mid v \in B_{M^{vis}}\}$

1  $Xps_{global} \leftarrow \bigcup_{v \in B_{M^{vis}}} Xps_v;$ $\quad$ // Collect
   unique xps
2  **foreach** $v \in B_{M^{vis}}$ **do**
3  $\quad Xps'_v \leftarrow Xps_v;$
4  $\quad$ **foreach** $\mathcal{X} \in Xps_{global}$ **do**
5  $\quad\quad$ **if** $\mathcal{X} \notin Xps'_v$ **then**
6  $\quad\quad\quad$ **if** $\texttt{WeakConXp}(\mathcal{X}, M^{vis}, v)$ **then**
7  $\quad\quad\quad\quad \mathcal{X} \leftarrow \texttt{ShrinkXp}(\mathcal{X}, M^{vis}, v);$
8  $\quad\quad\quad\quad Xps'_v \leftarrow Xps'_v \cup \{\mathcal{X}\};$

9  **return** $\{Xps'_v \mid v \in B_{M^{vis}}\};$

---

prunes the search tree wherever possible—if a set of concepts $\mathcal{X}$ is confirmed as an explanation, then the algorithm never checks if any extension of $\mathcal{X}$ is an explanation. The same algorithm is used for enumerating ConAXps and ConCXps with the appropriate checks being used in lines 2 and 9.

**XpEnum.** We adapt the XpEnum algorithm of Ignatiev et al. (2020b), which exploits the hitting set duality between AXps and CXps to simultaneously enumerate both types of explanations. The algorithm maintains the sets of ConAXps and ConCXps found so far. At each iteration, it computes a minimal hitting set of the found ConCXps (subject to not being a superset of any found ConAXp) as a candidate. If the candidate passes the WeakConAXp check—which, under monotonicity, reduces to a single oracle call—it is shrunk to a ConAXp. Otherwise, its complement is a WeakConCXp and is shrunk to a ConCXp. The newly found explanation is added to the corresponding collection, and the process repeats until no new candidates exist. Full details are in Appendix B.

**XpSatEnum.** Algorithm 2 describes a complementary procedure designed to address the incompleteness inherent in the previously described budget-bounded enumeration algorithms. Given that computing all ConAXps and ConCXps for a single image is often infeasible, we prioritize the discovery of those that occur frequently across images in a behavior $B_{M^{vis}}$. The algorithm takes as input a set of images $B_{M^{vis}}$ that corresponds to a behavior, together with initial per-image (contrastive or abductive) concept-based explanations $\{Xps_v \mid v \in B_{M^{vis}}\}$ produced by any budget-bounded enumerator (e.g., NaiveEnum or XpEnum) It collects all unique explanations $Xps_{global} = \bigcup_{v \in B_{M^{vis}}} Xps_v$ and then, for each image $v \in B_{M^{vis}}$, checks via $\texttt{WeakConXp}$ whether each explanation $\mathcal{X} \in Xps_{global}$ not already in $Xps_v$ is also valid for $v$. If so, the explanation is shrunk to a $\texttt{ConXp}$ (details of the $\texttt{ShrinkXp}$ procedure are in Appendix B). This cross-image saturation maximizes the overlap of the final explanation sets across images. The same algorithm is used for saturating both ConAXps and ConCXps with the appropriate checks. This method leverages the empirical observation that a minority of explanations are usually representative of most instances of a given behavior.

## 5  Experiments

In this section, we investigate the following research questions:

Table 1: Model architectures and training configurations. Summary of the five vision models evaluated, including the datasets used for pre-training and fine-tuning, and vocabulary size used to obtain explanations.

| Model | Backbone | Pre-trained | Fine-tuned | Vocab. size |
|-------|----------|-------------|------------|-------------|
| $M_1$ | CLIP | LAION400M | *zero-shot* | 150 |
| $M_2$ | ResNet18 | ImageNet | RIVAL10 | 300 |
| $M_3$ | ResNet18 | SSL4EO-RGB | EuroSAT | 300 |
| $M_4$ | VGG19 | ImageNet | RIVAL10 | 200 |
| $M_5$ | VGG19 | ImageNet | EuroSAT | 200 |

**RQ1.** (Generalizability) To what extent do explanations generalize over samples of the same behavior?

**RQ2.** (Coverage) Can a small subset of explanations cover the majority of images in a behavior?

**RQ3.** (Parsimony) Do short explanations have more coverage?

**RQ4.** (Plausibility) To what extent do models rely on relevant versus irrelevant concepts?

**RQ5.** (Efficiency) How does the choice of the enumeration and erasure algorithms affect the end-to-end computation time?

## 5.1 Empirical Setup

**Models.** We use CLIP Radford et al. (2021) as our reference Vision-Language model. We explore the vision models ResNet18 He et al. (2015), VGG19 Simonyan & Zisserman (2015), and CLIP itself as a zero-shot classifier. **Datasets.** All experimental evaluations are conducted on RIVAL10 Moayeri et al. (2022), which provides 10 classes from ImageNet about animals and objects (2000 images per category), and EuroSAT Helber et al. (2019), which consists of satellite images across 10 distinct geographical categories (2700 images per category). **Transfer Learning.** Except for zero-shot CLIP, each model had its final linear layer fine-tuned on the respective dataset before experimentation. We utilized VGG and ResNet weights pretrained on ImageNet for the RIVAL10 experiments. For EuroSAT, the ResNet model was initialized with weights pretrained on the SSL4EO-S12 Wang et al. (2023) dataset. **Vocabularies.** We use concept vocabularies pruned from a task-agnostic base vocabulary as described in Appendix E. **Behaviors.** For each model $M^{vis} \in \{M_1, \ldots, M_5\}$ in Table 1, we randomly choose one behavior of each type: misclassification and correct classification. Following Definition 3.7, we denote misclassification and correct classification behaviors as $B_{M^{vis}}(k_i, k_j)$ and $B_{M^{vis}}(k_i)$, respectively, where $k_i, k_j \in \mathcal{K}$ are classes defined in the RIVAL10 and EuroSAT datasets. For each behavior, we only consider the subset of images for which at least the trivial explanation (i.e., the one involving all concepts in the vocabulary) can be obtained, irrespective of which of the three erasure algorithms is used. When the number of such images in a behavior exceeds 700, we select a random subset of 700 images for computing explanations. Since the models being analyzed have high accuracies ($> 85\%$), the number of images per misclassification behavior typically consist of fewer than 100 images. Note that, for the VGG models, the subset of images with trivial explanation is often empty due to SpLiCE, so we drop SpLiCE as an erasure method for $M_4$ and $M_5$. With SpLiCE, the embedding can change even if no concepts are erased since it first sparsely decomposes the original embedding and then computes a new embedding from this sparse decomposition. For VGG, the new embeddings often cause a change in the prediction which rules out the trivial explanation. **Hyperparameters.** For the erasure methods, we use the following hyperparameters: tolerance ($\epsilon = 10^{-6}$) and $\ell_1$ regularization ($\lambda = 0.01$) on SpLiCE, and training sample size of 500 for LEACE to learn the affine transformation. For the explanation enumerators, we bound NaiveEnum's depth ($K = 2$) and XpEnum's iterations to 250. For XpSatEnum, the initial set of explanations are those computed by XpEnum. We set a 10-hour timeout for each run of an explanation enumerator.

Table 2: Average quantity of ConXps per image and its standard deviation for the model behaviors we analyze.

| Behavior | | Ortho | | | SpLiCE | | | LEACE | | |
|---|---|---|---|---|---|---|---|---|---|---|
| | | XpEnum | XpSatEn. | NaiveEn. | XpEnum | XpSatEn. | NaiveEn. | XpEnum | XpSatEn. | NaiveEn. |
| $B_{M_2}$(Car) | ConAXps | 42.5±14.1 | 547.7±36.8 | 216.2±27.2 | 31.6±1.8 | 91.6±2.7 | 0.0±2.8 | - | - | - |
| | ConCXps | 8.0±0.1 | 1601.7±4.2 | 3.5±2.7 | 16.4±7.3 | 56.2±9.1 | 10.4±35.5 | 3.0±0.9 | 167.3±7.1 | 258.6±50.1 |
| $B_{M_2}$(Truck,¬Truck) | ConAXps | 28.6±0.9 | 87.7±2.6 | 614.4±2.0 | 37.0±0.2 | 45.2±0.3 | - | - | - | - |
| | ConCXps | 7.1±0.2 | 45.8±1.4 | 400.9±1.0 | 17.8±1.1 | 25.9±1.4 | 43.1±2.3 | 10.0±0.8 | 52.2±1.4 | 420.9±2.2 |
| $B_{M_4}$(Cat) | ConAXps | 62.6±2.2 | 713.6±6.6 | 167.0±9.6 | - | - | - | - | - | - |
| | ConCXps | 7.1±0.1 | 336.8±2.4 | 48.1±3.6 | - | - | - | 10.0±5.0 | 80.9±11.2 | 792.9±17.4 |
| $B_{M_5}$(Lake,Pasture) | ConAXps | 194.9±1.5 | 10233.8±17.9 | 3.8±21.0 | - | - | - | - | - | - |
| | ConCXps | 51.6±2.1 | 665.0±13.9 | 464.7±18.8 | - | - | - | 10.0±19.2 | 36.6±34.7 | 1294.4±38.8 |

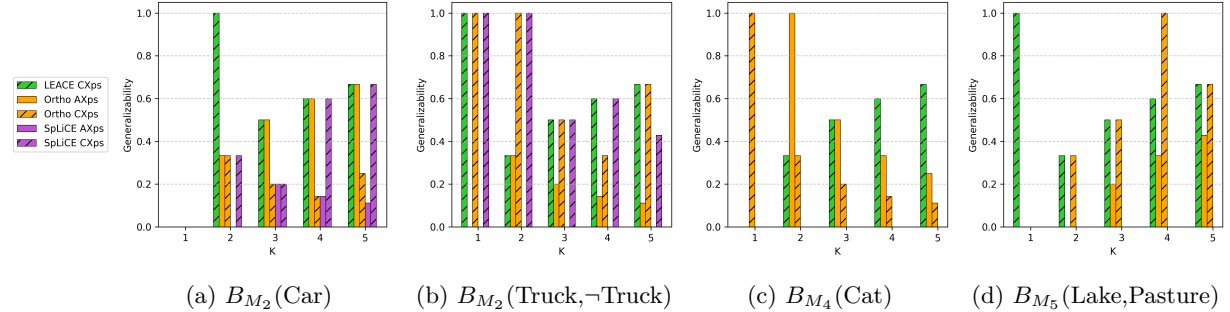

(a) $B_{M_2}$(Car)  (b) $B_{M_2}$(Truck,¬Truck)  (c) $B_{M_4}$(Cat)  (d) $B_{M_5}$(Lake,Pasture)

Figure 4: Generalizability@K for the model behaviors we analyze. See Section 5.1 and Definition 3.7 for details about these behaviors and their notation. More results in Appendix G.1.

## 5.2 Empirical Results

Our empirical evaluation consists of both quantitative and qualitative assessments. For each triple of (model, dataset, behavior), we compute explanations using all nine combinations of our explanation enumeration and erasure algorithms. We report the quantitative (Section 5.2) and qualitative results (Section G.8) on three models ($M_2$, $M_3$, and $M_4$) and a total of four behaviors in the main body of the paper. Additional empirical results are available in Appendix G.

Table 2 shows the number of explanations computed per combination of enumeration and erasure algorithms for the models and behaviors we analyze. Cases where no explanations were found within the budget are indicated with a dash (-). Note the difficulty in finding LEACE ConAXps. LEACE requires expensive linear algebraic calculations for every combination of concepts to be erased. When paired with XPEnum, it succeeds in finding ConCXps but fails to find ConAXps since XPEnum tends to compute only ConCXps in initial iterations before timing out. When LEACE is paired with NaiveEnum, while the search depth limited to $K \leq 2$ is sufficient to find ConCXps it proves insufficient for ConAXps. We also see that the VGG19 models ($M_4$,$M_5$) have no explanations when using SpLiCE since we drop it as an erasure method for these models for reasons stated in Section 5.1. In the remainder of this section, unless otherwise stated, for a (model, dataset, behavior) triple, we take the union of the set of abductive (or contrastive) explanations computed by three enumeration algorithms for each erasure method when reporting results to aid comprehension.

### 5.2.1 RQ1: Generalizability

An explanation is *generalizable* if it is applicable across different samples of the same behavior. A general explanation captures the essence of why the model exhibits a specific behavior as opposed to simply explaining model output on a single image. To evaluate the generalizability of our explanations, we define the following metric.

**Definition 5.1** (Generalizability@K)**.** Given two disjoint subsets $B_{M^{vis}}^{trn}$ and $B_{M^{vis}}^{tst}$ of a behavior $B_{M^{vis}}$, and corresponding concept-based explanations of these behavior subsets, $(h_A^{trn}, h_C^{trn})$ and $(h_A^{tst}, h_C^{tst})$ (where $h_A$ and $h_C$ are maps from signed ConAXps and ConCXps to counts as described in Definition 3.9), Generalizability@K is defined as:

$$\text{Gen@K}(B_{M^{vis}}^{trn}, B_{M^{vis}}^{tst}) := \left( \frac{|TopK(h_A^{trn}) \cap TopK(h_A^{tst})|}{|TopK(h_A^{trn}) \cup TopK(h_A^{tst})|}, \frac{|TopK(h_C^{trn}) \cap TopK(h_C^{tst})|}{|TopK(h_C^{trn}) \cup TopK(h_C^{tst})|} \right)$$

where $TopK(h)$ returns the top $k$ most frequent explanations in the map $h$.

Generalizability@K is evaluated by randomly partitioning a behavior's images into two disjoint subsets to compare their most frequent explanations. We use the Intersection over Union (IoU) of the top k most frequent explanations for the two sets as a measure of generalization of explanations across samples. An IoU of 1 indicates that the most frequent concept-based explanations are perfectly aligned between the two behavior subsets, suggesting that those explanations represent a generalized, universal explanation of that behavior. In contrast, an IoU of 0 implies that the most frequent explanations are entirely sample-dependent, indicating a failure in finding a generalized explanation for that behavior. Given that the set of possible concept-based explanations consists of thousands of unique elements, the probability of obtaining a high IoU by chance is statistically negligible. Therefore, any significant explanation overlap between the two disjoint sets provides strong evidence for the high quality and generalizability of the explanations.

Figure 4 shows Gen@K,K$\in [1, 5]$ for models $M_2$ and $M_4$ on one correct classification and misclassification behavior each. Results for other models are available in Appendix G.1. While increasing $K$ initially improves IoU by including more common explanations, we expect generalizability to decrease at much higher values (e.g., in the hundreds) as the sets begin to include low-frequency, sample-specific explanations. Such high $K$ values are impractical for human analysts, so we chose $K = 5$ as a reasonable upper bound. Similar generalizability values are observed for both correct and incorrect classifications in the behaviors we studied.

Generalizability values are highly sensitive to explanation length; longer explanations tend to report lower values because IoU treats sets that differ by even a single concept as distinct. While the choice of erasure algorithm generally has a small impact, the LEACE and NaiveEnum combination consistently provides high IoU values, see Appendix G.1 for more results.

> **Finding:** The most frequent ConXps remain stable across disjoint samples of behaviors in most cases, which is statistically improbable by chance; suggesting that ConXps are able to identify general, universal explanations of behaviors in vision models rather than just isolated observations.

### 5.2.2 RQ2: Coverage

For the datasets and behaviors we have explored, the total number of unique explanations can reach hundreds or thousands, exceeding the cognitive capacity of a human analyst. Selecting a small subset of explanations that applies to the majority of images in a behavior is essential for practical utility. A small set enables an analyst to prioritize highly-relevant concept patterns rather than accounting for every case. We define the next metric to quantify this balance of coverage and subset size as follows.

**Definition 5.2** (Maximum Coverage@K)**.** Given a behavior $B_{M^{vis}}$, model $M^{vis}$, concept vocabulary $\mathcal{C}$, and an explanation enumerator $\mathcal{E}$, let $\widehat{\mathcal{X}ps}_A$ be the set of all unique signed ConAXps (i.e., $\widehat{\mathcal{X}ps}_A = \{(\mathcal{X}, smap) \mid v \in B_{M^{vis}} \ \wedge \ \mathcal{X} \in \pi_1(\mathcal{E}(M^{vis}, v, \mathcal{C})) \ \wedge \ smap = sgn(\mathcal{X}, M^{vis}, v)\}$) found by the enumerator for the behavior. $\widehat{\mathcal{X}ps}_C$ is the set of all unique signed ConCXps defined analogously. We define Maximum Coverage@K as the maximum number of images that can be explained by K signed ConAXps (similarly, for ConCXps).

$$\text{MaxCov@K}(B_{M^{vis}}) := \left( \max_{S_A} \left| \bigcup_{\mathcal{X}_i \in S_A} B^{\mathcal{X}_i} \right|, \max_{S_C} \left| \bigcup_{\mathcal{X}_i \in S_C} B^{\mathcal{X}_i} \right| \right) \ \text{s.t.} \ S_A \subseteq \widehat{\mathcal{X}ps}_A, S_C \subseteq \widehat{\mathcal{X}ps}_C, |S_A| = |S_C| = K$$

where $B^{\mathcal{X}_i} \subseteq B_{M^{vis}}$ is the set of images for which $\mathcal{X}_i$ is a signed explanation.

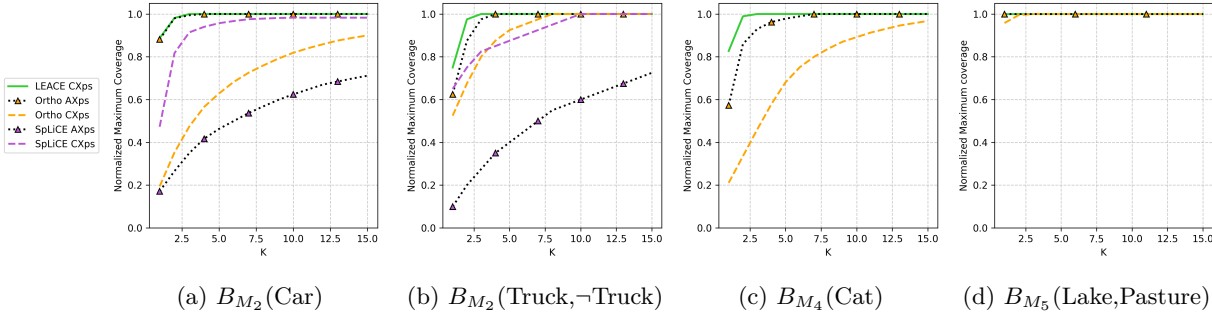

Figure 5: Maximum Coverage@K for the model behaviors we analyze. More results in Appendix G.2.

This is an instance of the *Maximum Coverage Problem*, an NP-hard problem where given a set of subsets of a set—in our case, subsets $\{B^{\mathcal{X}_1}, B^{\mathcal{X}_2}, \ldots B^{\mathcal{X}_n}\}$ of set $B_{M^{vis}}$—and an integer $K$, the goal is to select at most $K$ subsets that maximize the total number of unique elements covered. We use a greedy strategy that iteratively selects the next explanation $\mathcal{X}_i$ with the highest marginal gain in image coverage; algorithm detailed in Appendix F.

As shown in Figure 5, we report the Normalized Maximum Coverage@$K$, where the number of images explained by the explanations in MaxCov@K is divided by the total number of images in the behavior of interest. A value of 1 indicates the selected subset of explanations covers the entire behavior. Our experiment shows that only a small subset of explanations is required to account for the majority of images within a behavior: across all four behaviors explored, at least one erasure algorithm achieves a normalized maximum coverage value of 1 with $K \leq 5$ explanations, while runner-up candidates reliably do so with $K \leq 10$; more results at Appendix G.2. Considering that the search space contains thousands of unique explanations, the ability to achieve full coverage with 10 or fewer explanations demonstrates the practical utility of our approach. Notably, no single erasure algorithm consistently outperforms the others; instead, their performance appears to be behavior-dependent. This empirical observation points to an opportunity for future research.

**Individual Coverage.** Another aspect of interest to a human analyst is the explanatory power of individual explanations, measured by the number of images they individually explain. We quantify this by calculating the **Individual Coverage** of each explanation, defined as the number of images to which a specific explanation applies divided by the total number of images in the behavior of interest. A value of 1 implies that an explanation accounts for the entire behavior. Figure 6 shows these Individual Coverages in descending order. The empirical results show that the distribution of individual coverages is highly skewed across all behaviors. A small number of explanations report high individual coverages, while a large number of explanations apply only to a few instances. This result confirms that our approach effectively separates general concept patterns from niche cases, allowing an analyst to focus on the most influential explanations. Others Sokol & Flach (2020) have proposed similar metrics for evaluating the quality of computed explanations.

**Mixed Behaviors.** Inspired by Dunlap et al. (2024), we evaluate the impact of mixing behaviors on the individual coverages of explanations. Given a collection of $n$ behaviors $B^{(1)}, B^{(2)}, \ldots, B^{(n)}$, we define a **Mixed Behavior Set** as their union $B^{mixed} = \bigcup_{i=1}^{n} B^{(i)}$ and then measure individual coverage of explanations on the mixed set. Our results show that while a specific explanation may account for a large portion of $B^{(i)}$, its individual coverage decreases significantly when calculated over $B^{mixed}$. As shown in Figure 7, the resulting curve is much less skewed than in Figure 6, which indicates that the explanations report lower individual coverages across the mixed set. The observed drop in individual explanatory power supports the intuition that the identified explanations are characteristic of the behavior of interest and do not generalize to unrelated samples by chance.

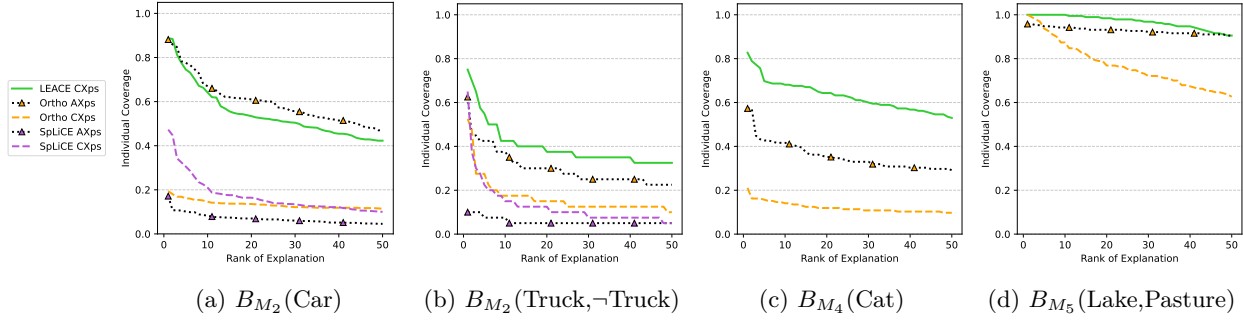

Figure 6: Individual coverages for the model behaviors we analyze. While the top-ranked explanations show broad applicability, a long flat tail in the plot suggests many explanations cover only a few images, e.g., corner cases. This trend is consistent across different models, behaviors, and erasure algorithms.

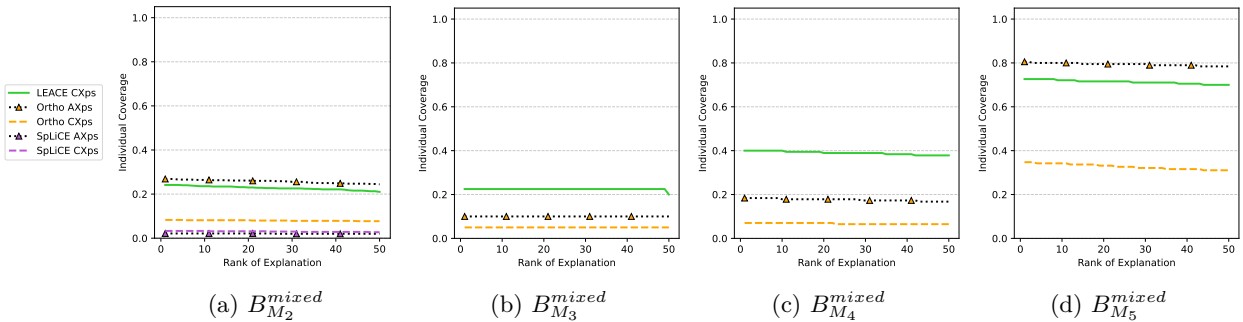

Figure 7: Individual coverage on Mixed Behavior Sets for the models we analyze.

> **Finding:** Despite obtaining thousands of unique ConXps per image, (i) a small subset of $K \leq 10$ explanations is sufficient to cover all images in most explored behaviors, and (ii) their individual coverage is highly skewed, implying that only a small subset of explanations requires human attention.

### 5.2.3 RQ3: Parsimony

Following Occam's razor Rasmussen & Ghahramani (2000), or the law of parsimony, which recommends searching for explanations constructed with the smallest possible set of elements, Sokol & Flach (2020) state that explanations should be selective and succinct enough to avoid overwhelming the explainee with unnecessary information, i.e., fill in the most gaps with the fewest arguments.

We analyze the association between explanation size, defined as the number of included concepts, and individual coverage. Figure 8 shows this distribution using a violin plot where the width at a given y-axis value represents the proportion of explanations reporting that value of individual coverage. The results in Figure 8 reveal a trend where smaller explanations of $L \leq 2$, where $L$ is the number of concepts in an explanation, report higher individual coverage than larger explanations of $L \geq 3$ concepts. This suggests that parsimonious explanations are more likely to capture the broad, systematic patterns of a model's behavior.

> **Finding:** Short ConXps consistently report higher individual coverage than larger ones across evaluated models, datasets, and behaviors. This suggests that searching for large ConXps is unnecessary, as increasing the number of concepts is negatively correlated with explanations' individual coverage.

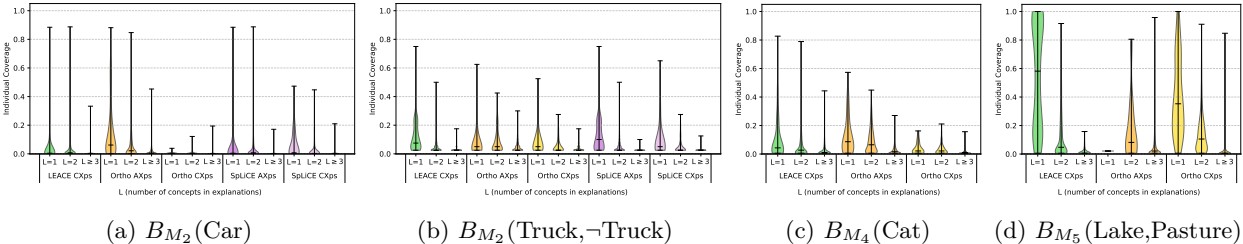

(a) $B_{M_2}$(Car)  (b) $B_{M_2}$(Truck,¬Truck)  (c) $B_{M_4}$(Cat)  (d) $B_{M_5}$(Lake,Pasture)

Figure 8: Explanation Size vs. Individual Coverage for the model behaviors we analyze.

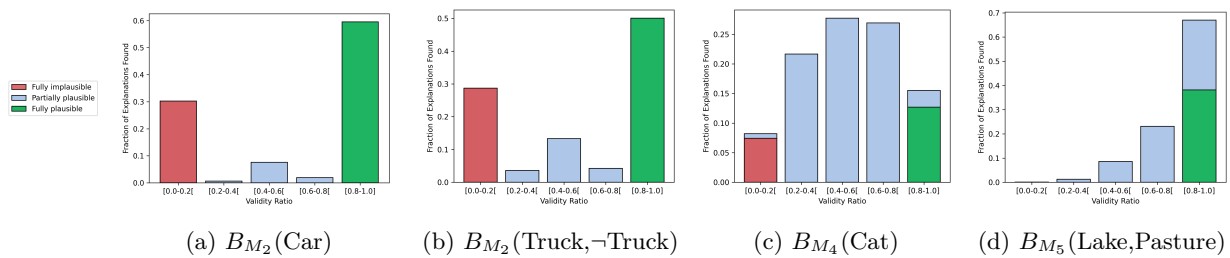

(a) $B_{M_2}$(Car)  (b) $B_{M_2}$(Truck,¬Truck)  (c) $B_{M_4}$(Cat)  (d) $B_{M_5}$(Lake,Pasture)

Figure 9: Fraction of the total number of explanations that are fully plausible (green), partially plausible (light blue), and fully implausible (red) explanations across the model behaviors we analyze. *Validity Ratio* is the proportion of the number of valid concepts to total number of concepts in a explanation. Each bar shows the fraction of total explanations where the validity ratio is in the specified range.

### 5.2.4   RQ4: Plausibility

We categorize the plausibility of each explanation by evaluating if its constituent concepts are *relevant* to the model behavior. A signed explanation is *fully plausible* if all its positive concepts are relevant and all its negative concepts are irrelevant for the behavior. In contrast, an explanation is fully implausible if all positive concepts are irrelevant and negative concepts are relevant. Measuring the fraction of the total number of explanations that are plausible, partially plausible, and implausible acts as a lens to determine if the vision model's decisions are based on relevant concepts or on spurious correlations. Following the *LLM-as-a-judge* strategy Li et al. (2025), we utilized Gemini 3 Pro  Google DeepMind (2026) to classify each concept in our base vocabulary as relevant or irrelevant for the model behavior under consideration. Please refer to Appendix H for prompt templates used.

As shown in Figure 9, the evaluated vision models show a tendency towards plausible explanations over implausible ones. However, a significant proportion of explanations are only partially plausible, indicating that many decisions are based on a mix of relevant and spurious concepts. These spurious concepts often represent features that are highly correlated with a behavior but are not causal, such as the presence of water in images of ships. The existence of partially plausible explanations suggests that while models prioritize relevant features, they still use non-essential contextual information in many of their decisions.

The following are three examples of behaviors with explanations that, despite being among the top five in individual coverage, contain irrelevant concepts: $B_{M_1}$(Annual Crop) reports {Fairy(+)} and {Coal(+)}, $B_{M_1}$(Ship) reports {Boat(+)∧Sausage(+)∧Wreck(+)∧Sea(+)}, and $B_{M_1}$(Sea Lake, Forest) reports {Smoke(+)∧Rat(+)∧Duck(+)} and {Fog(+)∧Rat(+)∧Duck(+)}. More results in Appendix G.4. Among the models and behaviors studied, CLIP generally provides a higher percentage of plausible explanations, followed closely by ResNet. In contrast, VGG reports the lowest plausibility and a higher frequency of spurious correlations. This difference may be attributed to the performance of linear maps, see Appendix D.1, rather than the models' inherent logic; this empirical observation opens an interesting future study.

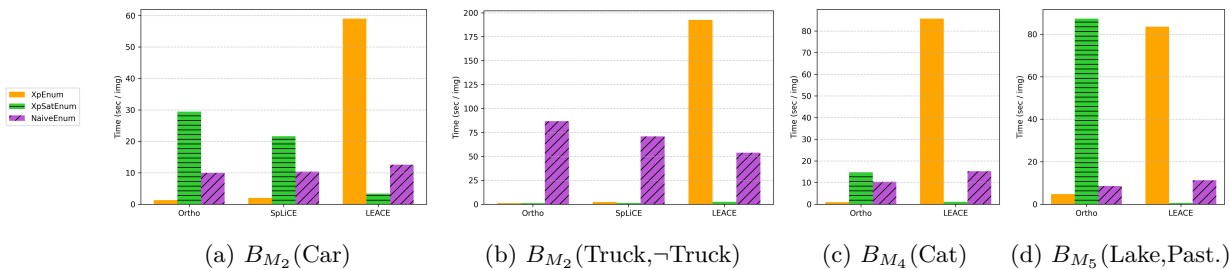

(a) $B_{M_2}(\text{Car})$   (b) $B_{M_2}(\text{Truck},\neg\text{Truck})$   (c) $B_{M_4}(\text{Cat})$  (d) $B_{M_5}(\text{Lake,Past.})$

Figure 10: Compute time per image, measured in seconds, for the model behaviors we analyze. See Figure 17 on Appendix G.5 for compute times on additional behaviors.

> **Finding:** ConXps can be used to detect spurious reasoning in vision models by identifying cases where a behavior relies on irrelevant concepts. The models and behaviors we analyze largely rely on relevant concepts, though the presence of partially plausible explanations is not negligible.

### 5.2.5 RQ5: Efficiency

We compare the time required per combination of a erasure algorithm with an explanation enumeration algorithm to compute explanations per image. We refer to each such combination as a *configuration*. Figure 10 shows the computation times for each algorithm combination. XpEnum is considerably faster with Ortho and SpLiCE, but is the slowest by far when used with LEACE. XpSatEnum times vary based on the number of explanations found by XpEnum. NaiveEnum is consistently fast across all combinations, though not the fastest overall. This stability makes NaiveEnum a reliable choice across different erasure choices. Detailed discussion is available in Appendix G.5.

> **Finding:** Although formal explanation methods often struggle to scale to large models and datasets, our approach offers a reasonable trade-off between scalability and explanation quality.

## 6 Related Work

Research on explaining machine learning behavior spans a wide range of techniques. Approaches for obtaining *formal explanations* use logical encodings, abductive reasoning, and SAT/MaxSAT formulations to derive minimal sets of input features that guarantee a model's prediction. This enables counterfactual, adversarial, and causal analyses with strong formal guarantees Ignatiev et al. (2019a; 2020a); Shao et al. (2024); Ignatiev et al. (2019b); Reddy et al. (2024); Ignatiev & Marques-Silva (2021); Izza & Marques-Silva (2021); Ignatiev et al. (2022); Audemard et al. (2023a;b; 2024); Bassan & Katz (2023); Yu et al. (2024). However, explanations expressed over raw input variables (such as pixels for visual modality), are difficult to interpret. *Mechanistic interpretability* methods instead employ interventions such as activation patching, causal tracing, and mediation analysis on internal model representations to identify components responsible for specific behaviors Vig et al. (2020); Meng et al. (2022); Conmy et al. (2023); Zhang & Nanda (2024); Ahmad et al. (2024); Xia et al. (2021). This provides insights into how models compute outputs, however, they lack semantic meaning since they are in terms of neurons, attention heads, or latent subspaces.

*Concept-based interpretability* addresses this gap by explaining behavior in terms of human-understandable concepts using techniques such as concept activation vectors, concept erasure, amnesic probing, and counterfactual representation interventions Kim et al. (2018); Belrose et al. (2023); Elazar et al. (2021); Ravfogel et al. (2021). Vision-language models enable grounding explanations in textual concept spaces Kim et al. (2023); Bhalla et al. (2024); Gandelsman et al. (2024); Balasubramanian et al. (2024); Papadimitriou et al. (2025). Recent work leverages VLMs to interpret, debug, and monitor vision models by mapping their respective representation spaces Hu et al. (2025). Closely related are methods performing formal reasoning

over structured semantic variables derived from programmatic scene descriptions or causal abstractions Kim et al. (2020); Rajendran et al. (2024); Bassan et al. (2023).

Our work bridges formal explanation methods and concept-based interpretability by enabling formally grounded abductive and contrastive explanations over semantically meaningful concepts. Our novelty lies in combining semantic grounding (leveraging VLMs) with guarantees of minimality and causal relevance.

## 7 Conclusion

We presented the new notion of concept-based abductive and contrastive explanations. These explanations provide causal explanations for model behaviors in terms of minimal sets of sufficient or necessary concepts. Causality is determined by erasing the concepts from internal model representations. We presented three algorithms, parametric in the concept erasure procedure, for enumerating such explanations. Our algorithms assume that the model head is monotonic with respect to concepts. We also presented a new concept erasure procedure based on the intuition that the representation, after erasure, should be orthogonal to each of the erased concepts while being as similar to original representation as possible. For the models and behaviors we analyze, short, human-understandable, explanations tend to be most explanatory. The inferred explanations also generalize to unseen images where the model exhibits the same behavior. As future work, we plan to apply these methods to LLMs and VLMs where model behaviors are characterized over sequences of generated texts instead of categorical predictions. We expect this richer behavior to necessitate adding a temporal aspect to our concept-based explanations to account for the auto-regressive nature of these models.

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

# A Proof of Theorem 3.1

**Theorem 3.1**(Concept-based Explanations Under Monotonicity) Given a vision model $M^{vis}$, if we assume that $M^{head}$ is monotonic, then the definitions of weak concept-based abductive and contrastive explanations can be simplified as follows:

$$\texttt{WeakConAXp}(\mathcal{X}, M^{vis}, v, k) := \forall (b \in \mathbb{B}). \bigwedge_{j \in \mathcal{X}} (b_j = 1) \bigwedge_{j \notin \mathcal{X}} (b_j = 0) \to (M^{head}(erase(M^{enc}(v), b)) = k)$$

$$\texttt{WeakConCXp}(\mathcal{X}, M^{vis}, v, k) := \exists (b \in \mathbb{B}). \bigwedge_{j \notin \mathcal{X}} (b_j = 1) \bigwedge_{j \in \mathcal{X}} (b_j = 0) \wedge (M^{head}(erase(M^{enc}(v), b)) \neq k)$$

*Proof.* We prove each simplification separately.

`WeakConAXp.` We show equivalence between the original and simplified definitions.

($\Rightarrow$) (original implies simplified.) The simplified definition is a special case of the original (it restricts $b_j = 0$ for $j \notin \mathcal{X}$), so the original immediately implies the simplified.

($\Leftarrow$) (simplified implies original.) Suppose the simplified definition holds, i.e., for the assignment $b^0$ defined by $b_j^0 = 1$ for $j \in \mathcal{X}$ and $b_j^0 = 0$ for $j \notin \mathcal{X}$, we have $M^{head}(erase(M^{enc}(v), b^0)) = k$. Now let $b$ be any assignment satisfying $b_j = 1$ for all $j \in \mathcal{X}$. The assignment $b$ can be obtained from $b^0$ by setting some indices $j \notin \mathcal{X}$ from 0 to 1. By repeated application of the monotonicity assumption, each such change preserves the prediction $k$. Therefore $M^{head}(erase(M^{enc}(v), b)) = k$.

`WeakConCXp.` We again show equivalence.

($\Leftarrow$) (simplified implies original.) The simplified definition provides a concrete witness $b^0$ (with $b_j^0 = 1$ for $j \notin \mathcal{X}$ and $b_j^0 = 0$ for $j \in \mathcal{X}$) satisfying $M^{head}(erase(M^{enc}(v), b^0)) \neq k$. This $b^0$ also satisfies the conditions of the original definition, so the original holds.

($\Rightarrow$) (original implies simplified.) Suppose the original definition holds, i.e., there exists some $b^*$ with $b_j^* = 1$ for $j \notin \mathcal{X}$ and $M^{head}(erase(M^{enc}(v), b^*)) \neq k$. Define $b^0$ as above with $b_j^0 = 0$ for $j \in \mathcal{X}$. Note that $b^*$ can be obtained from $b^0$ by setting some indices $j \in \mathcal{X}$ from 0 to 1. The contrapositive of the monotonicity assumption states that $M^{head}(erase(M^{enc}(v), b[j \mapsto 1])) \neq k \Rightarrow M^{head}(erase(M^{enc}(v), b)) \neq k$. Applying this repeatedly to walk from $b^*$ back to $b^0$ (flipping the relevant indices $j \in \mathcal{X}$ from 1 to 0), the assumption $M^{head}(erase(M^{enc}(v), b^*)) \neq k$ yields $M^{head}(erase(M^{enc}(v), b^0)) \neq k$, establishing the simplified definition. $\square$

## A.1 Empirical Validation of Head Monotonicity

This section provides empirical evidence supporting our assumption that vision model heads are monotonic in practice. Following our test detailed in Algorithm 3, we evaluated the $N = 50$ ConAXps and ConCXps with the highest individual coverage, as these represent the most relevant explanations for human analysts. We repeated this test for ten behaviors, five of each type, spanning all models and datasets explored. We considered ConXps obtained via all three explanation enumerators, and the Ortho erasure algorithm. For each ConXp, we randomly chose $L = 5$ images and iteratively included $m = 10$ random concepts, one by one. On each iteration, we performed oracle calls (`WeakConAXp` and `WeakConCXp`) to check if the *'relaxed'* concept sets remained valid weak explanations. Finally, we obtained the average proportion of cases where the assumption holds; reported in output variables $O_A$ and $O_C$ as percentages.

**Why is this a monotonicity check?** The procedure of Algorithm 3 directly probes the monotonicity assumption rather than a proxy for it. A verified ConAXp $\mathcal{X}_A$ certifies that $M^{head}(erase(M^{enc}(v), b^0)) = k$ at the assignment $b^0$ defined by $b_j^0 = 1$ for $j \in \mathcal{X}_A$ and $b_j^0 = 0$ for $j \notin \mathcal{X}_A$. Augmenting $\mathcal{X}_A$ with a random concept $c$ produces $\mathcal{X}_A' = \mathcal{X}_A \cup \{c\}$, and the simplified `WeakConAXp` check on $\mathcal{X}_A'$ evaluates $M^{head}$ at $b^0[c \mapsto 1]$ which is exactly the right-hand side of the monotonicity implication for index $j = c$. Hence $O_A$ reports the empirical rate at which the forward direction of monotonicity holds, starting from a verified

---

**Algorithm 3:** Empirical Monotonicity Test

---

**Input:** Vision Model $M^{vis} = M^{head} \circ M^{enc}$, Behavior $B_{M^{vis}}(k)$, ConXp sets $TopN(h_A)$ and
$TopN(h_C)$, Vocabulary $\mathcal{C}$, Image Count $L$, Iterations $m$

**Output:** Percentage of cases where (i) adding concepts does not change behavior ($O_A$); (ii) erasing
concepts does not change behavior ($O_C$)

**1** $O_A \leftarrow 0;\ O_A^* \leftarrow \mathbf{0} \in \mathbb{B}^{N \times L \times m}$;

**2 foreach** $\mathcal{X}_A \in TopN(h_A)$ **do**                // Original ConAXps (#changes = 0)

**3**  $V \leftarrow \{v \mid \text{ConAXp}(\mathcal{X}_A, M^{vis}, v, k)\}$ s.t. $|V| = L$;

**4**  **foreach** $v \in V$ **do**

**5**    $\mathcal{X}_A' \leftarrow \mathcal{X}_A$;

**6**    **for** $i = 1$ **to** $m$ **do**

**7**      $\text{con}_i \leftarrow \text{RandomPickOne}(\mathcal{C} \setminus \mathcal{X}_A')$;

**8**      $\mathcal{X}_A' \leftarrow \mathcal{X}_A' \cup \{\text{con}_i\}$ ;     // Relaxed WeakConAXps (#changes = i)

**9**      $O_A^*(n, l, i) \leftarrow \text{WeakConAXp}(\mathcal{X}_A', M^{vis}, v, k)$;

**10** $O_A \leftarrow \frac{1}{NLm} \sum_{n=1}^{N} \sum_{l=1}^{L} \sum_{i=1}^{m} O_A^*(n, l, i) * 100$;      // $O_C$ obtained similarly

---

Table 3: Monotonicity empirical test results and standard deviation (params: $N = 50$, $L = 5$, and $m = 10$).
Cases where not enough images were available to satisfy image count L are indicated with a dash (-).

| **Behavior** | $B_{M_1}$(Deer) | $B_{M_2}$(Car) | $B_{M_3}$(A.Crop) | $B_{M_4}$(Cat) | $B_{M_5}$(A.Crop) |
|---|---|---|---|---|---|
| $O_A$ | 87.80±0.32 | 72.20±0.46 | 70.88±0.46 | 75.32±0.44 | 98.86±0.10 |
| $O_C$ | 88.84±0.33 | 87.16±0.33 | 72.04±0.47 | 75.76±0.42 | 78.6±0.40 |

| **Behavior** | $B_{M_1}$(Equine,Car) | $B_{M_2}$(Tru,¬Tru) | $B_{M_3}$(River,A.Cr) | $B_{M_4}$(Deer,Car) | $B_{M_5}$(Lake,Past) |
|---|---|---|---|---|---|
| $O_A$ | - | 65.46±0.49 | 52.52±0.49 | - | 99.12±0.10 |
| $O_C$ | 92.32±0.28 | 93.93±0.23 | 89.96±0.30 | 83.40±0.36 | 79.64±0.40 |

ConAXp. Symmetrically, for a verified ConCXp $\mathcal{X}_C$ the canonical assignment has $b_j^0 = 1$ for $j \notin \mathcal{X}_C$ and $b_j^0 = 0$ for $j \in \mathcal{X}_C$; adding a concept $c$ to $\mathcal{X}_C$ flips $b_c^0$ from 1 to 0, and the `WeakConCXp` check then tests whether the prediction $\neq k$ is preserved under this flip, which is the contrapositive of monotonicity.

**Why these results justify our use of the assumption?** Table 3 shows that monotonicity is not strictly satisfied in either direction—both the forward rate $O_A$ and the contrapositive rate $O_C$ fall below 100% across most behaviors. The implications for our algorithms are asymmetric: while violations on the ConAXp side require empirical justification, violations on the ConCXp side do not affect algorithm correctness at all.

For ConAXps, the universal definition requires $M^{head}(erase(M^{enc}(v), b)) = k$ at every $b \in \mathbb{B}$ with $b_j = 1$ for $j \in \mathcal{X}_A$, while the simplified check probes only the canonical $b^0$ defined by $b_j^0 = 1$ for $j \in \mathcal{X}_A$ and $b_j^0 = 0$ for $j \notin \mathcal{X}_A$. Empirical evidence is therefore needed to argue that the universal claim holds for the explanations our algorithms report. Each iteration of Algorithm 3 extends $\mathcal{X}_A$ with random additional concepts and runs the simplified WeakConAXp check on the larger set. Mechanically, this evaluates $M^{head}$ at a new $b$ obtained from $b^0$ by setting up to ten additional 1-bits at indices outside $\mathcal{X}_A$. Since this $b$ still satisfies $b_j = 1$ for $j \in \mathcal{X}_A$, it falls within the set of $b$'s the universal WeakConAXp definition requires the prediction to equal $k$ at. The averaged $O_A$ rate is therefore an empirical estimate of the probability that the prediction is preserved at $b$'s drawn from this neighborhood of $b^0$, a probabilistic relaxation of the universal claim. The observed rates indicate that, while strict universality cannot be guaranteed, the universal property is empirically preserved for the great majority of $b$'s in this neighborhood, supporting, in a probabilistic sense, the simplified-to-universal lift our algorithms rely on.

For ConCXps, although Table 3 shows that the contrapositive direction of monotonicity is also imperfectly satisfied, this has no consequence for algorithm correctness. Theorem 3.1 simplifies the WeakConCXp definition under monotonicity by replacing the existential over $b \in \mathbb{B}$ with a check at the single canonical $b^0$.

While that simplification at the WeakConCXp level requires monotonicity, the corresponding simplification at the ConCXp level (i.e., for subset-minimal sets) does not, as the following theorem shows.

**Theorem A.1** (ConCXp Simplification Without Monotonicity). *Without requiring the monotonicity assumption of Theorem 3.1, the definition of concept-based contrastive explanation can be simplified as follows:*

$$\texttt{ConCXp}(\mathcal{X}, M^{vis}, v, k) := \texttt{WeakConCXp}_{simp}(\mathcal{X}, M^{vis}, v, k) \wedge \left( \forall \mathcal{X}' \subsetneq \mathcal{X}. \neg \texttt{WeakConCXp}_{simp}(\mathcal{X}', M^{vis}, v, k) \right)$$

*where $\texttt{WeakConCXp}_{simp}$ denotes the simplified WeakConCXp definition from Theorem 3.1. In other words, if a set is a ConCXp under the simplified WeakConCXp, it is also a ConCXp under the original WeakConCXp, and vice versa without needing monotonicity.*

*Proof.* We show the equivalence between the original ConCXp definition (using the original $\texttt{WeakConCXp}$) and the simplified one above. Both directions rest on the same observation: any witness $b$ for the original $\texttt{WeakConCXp}(\mathcal{X})$ that differs from the canonical $b^0$ (with $b_j^0 = 1$ for $j \notin \mathcal{X}$ and $b_j^0 = 0$ for $j \in \mathcal{X}$) yields a strictly smaller WeakConCXp $\mathcal{X}^* \subsetneq \mathcal{X}$, with the same $b$ acting as its canonical witness.

($\Rightarrow$) *Original implies simplified.* Suppose $\mathcal{X}$ is a ConCXp under the original definition. By $\texttt{WeakConCXp}(\mathcal{X})$, some $b$ with $b_j = 1$ for $j \notin \mathcal{X}$ satisfies $M^{head}(erase(M^{enc}(v), b)) \neq k$. Let $\mathcal{X}^* = \{j \mid b_j = 0\}$. Since $b_j = 1$ for all $j \notin \mathcal{X}$, every erased index lies in $\mathcal{X}$, so $\mathcal{X}^* \subseteq \mathcal{X}$. If $\mathcal{X}^* \subsetneq \mathcal{X}$, then $b$ also witnesses $\texttt{WeakConCXp}(\mathcal{X}^*)$ (it has $b_j = 1$ for all $j \notin \mathcal{X}^*$), contradicting subset-minimality of $\mathcal{X}$. Hence $\mathcal{X}^* = \mathcal{X}$, so $b = b^0$, and $\texttt{WeakConCXp}_{simp}(\mathcal{X})$ holds. For the second conjunct: if some $\mathcal{X}' \subsetneq \mathcal{X}$ satisfied $\texttt{WeakConCXp}_{simp}$, its canonical assignment would witness $\texttt{WeakConCXp}(\mathcal{X}')$, again contradicting subset-minimality.

($\Leftarrow$) *Simplified implies original.* Suppose the simplified definition holds. The first conjunct says $\texttt{WeakConCXp}_{simp}(\mathcal{X})$, so $b^0$ witnesses $\texttt{WeakConCXp}(\mathcal{X})$, making $\mathcal{X}$ a WeakConCXp. For subset-minimality under the original definition: suppose $\mathcal{X}' \subsetneq \mathcal{X}$ is a WeakConCXp, witnessed by some $b$. Let $\mathcal{X}^* = \{j \mid b_j = 0\}$. Since $b$ witnesses $\texttt{WeakConCXp}(\mathcal{X}')$, we have $b_j = 1$ for all $j \notin \mathcal{X}'$, so every erased index lies in $\mathcal{X}'$, giving $\mathcal{X}^* \subseteq \mathcal{X}' \subsetneq \mathcal{X}$. Moreover, $b$ is exactly the canonical $b^{0,*}$ for $\mathcal{X}^*$, since $b_j = 0$ precisely when $j \in \mathcal{X}^*$ and $b_j = 1$ otherwise. So $\texttt{WeakConCXp}_{simp}(\mathcal{X}^*)$ holds, contradicting the second conjunct of the simplified definition. Hence no $\mathcal{X}' \subsetneq \mathcal{X}$ is a WeakConCXp, so $\mathcal{X}$ is subset-minimal, i.e., a ConCXp. □

Theorem A.1 shows that the algorithms' use of simplified checks correctly characterizes ConCXps regardless of whether monotonicity holds. The empirical $O_C$ rates in Table 3 therefore complement $O_A$ in measuring monotonicity in both directions, but do not serve as evidence for algorithm correctness on the ConCXp side.

**Sampling design.** Each iteration of Algorithm 3 adds a random concept to $\mathcal{X}_A$ (or $\mathcal{X}_C$), producing a $b$ within a small Hamming distance of $b^0$. An alternative would be to sample $b$ uniformly from the set the relevant formal definition quantifies over. We prefer the biased design. For ConAXp, the formal definition's domain $\{b \in \mathbb{B} \mid b_j = 1 \text{ for } j \in \mathcal{X}_A\}$ has $2^{n-|\mathcal{X}_A|}$ elements (e.g., $\approx 2^{290}$ for $n = 300$, $|\mathcal{X}_A| = 10$), and a uniformly drawn $b$ has expected Hamming distance from $b^0$ of about $(n - |\mathcal{X}_A|)/2$. The typical uniform $b$ thus has roughly half the unconstrained concepts un-erased, putting the resulting embedding close to the model's original output where the prediction is $k$ by construction, regardless of monotonicity. Uniform sampling would therefore dilute the test with easy cases and inflate the reported rate. The interesting regime, where monotonicity violations are most likely to manifest, is near $b^0$ where the embedding is most perturbed from the original. The biased design concentrates on this regime and produces a more conservative estimate of the head's monotonicity.

# B    XpEnum Algorithm

We adapt the XpEnum algorithm of Ignatiev et al. (2020b) to enumerate concept-based formal explanations. The original algorithm exploits the hitting set duality between abductive and contrastive explanations (AXps are minimal hitting sets of CXps and vice-versa) to simultaneously enumerate both types of explanations. We adapt this to our concept-based setting, where the entailment checks reduce to single oracle calls under the monotonicity assumption (Theorem 3.1).

---

**Algorithm 4:** XpEnum: Enumeration of ConAXps and ConCXps

---

**Input:** Model $M^{vis}$, Image $v$, Prediction $k = M^{vis}(v)$, Vocabulary $\mathcal{C}$
**Output:** Sets of ConAXps and ConCXps

1  $AXps \leftarrow \emptyset$; $CXps \leftarrow \emptyset$;
2  **while** *true* **do**
3      $\mathcal{S} \leftarrow \text{FindMHS}(CXps, AXps)$ ;             // MHS of $CXps$ s.t. $AXps$; $\bot$ if none exists
4      **if** $\mathcal{S} = \bot$ **then return**;
5      **if** *WeakConAxp*$(\mathcal{S}, M^{vis}, v, k)$ **then**
6         $\mathcal{X} \leftarrow \text{ShrinkAXp}(\mathcal{S}, M^{vis}, v)$;
7         $AXps \leftarrow AXps \cup \{\mathcal{X}\}$;
8      **else**
9         $\mathcal{Y} \leftarrow \text{ShrinkCXp}(\mathcal{C} \setminus \mathcal{S}, M^{vis}, v)$;
10       $CXps \leftarrow CXps \cup \{\mathcal{Y}\}$;

---

Algorithm 4 presents the adapted procedure. The algorithm maintains two collections: $AXps$, the set of ConAXps found so far, and $CXps$, the set of ConCXps found so far. At each iteration, it computes a candidate $\mathcal{S}$ by finding a minimal hitting set of the ConCXps found so far, subject to not being a superset of any previously found ConAXp. This is performed by $\text{FindMHS}(CXps, AXps)$, which can be implemented using a MaxSAT solver Ignatiev et al. (2020b). If no such hitting set exists, all ConAXps and ConCXps have been enumerated and the algorithm terminates.

The candidate $\mathcal{S}$ is then tested via $\text{WeakConAxp}$: under the monotonicity assumption, this amounts to erasing all concepts not in $\mathcal{S}$ while keeping concepts in $\mathcal{S}$ at their original values, and checking whether the prediction is preserved, i.e., $M^{head}(erase(M^{enc}(v), b)) = k$ where $b_j = 1$ for $j \in \mathcal{S}$ and $b_j = 0$ for $j \notin \mathcal{S}$.

If $\mathcal{S}$ passes the check, it is a WeakConAXp and is shrunk to a subset-minimal ConAXp via $\text{ShrinkAXp}$. This is a standard deletion-based procedure: for each concept $c \in \mathcal{S}$, it checks whether $\mathcal{S} \setminus \{c\}$ is still a WeakConAXp; if so, $c$ is removed. The result is a ConAXp $\mathcal{X} \subseteq \mathcal{S}$.

If $\mathcal{S}$ fails the check, the complement $\mathcal{C} \setminus \mathcal{S}$ is guaranteed to be a WeakConCXp—erasing all concepts in $\mathcal{C} \setminus \mathcal{S}$ while keeping those in $\mathcal{S}$ changed the prediction. This set is then shrunk to a subset-minimal ConCXp via $\text{ShrinkCXp}$, which operates analogously: for each concept $c \in \mathcal{C} \setminus \mathcal{S}$, it checks whether the set without $c$ is still a WeakConCXp (i.e., erasing the remaining concepts still changes the prediction); if so, $c$ is removed.

Under the monotonicity assumption, each call to $\text{WeakConAxp}$ or $\text{WeakConCxp}$ (within the shrinking procedures) requires a single forward pass through the model, making the algorithm practical for vision models where each oracle call involves an erasure operation followed by a prediction.

## C Derivation of Ortho Erasure Procedure

### C.1 Setup and problem statement

Let $e \in \mathbb{R}^d$ be an embedding, and let $c_1, \ldots, c_n \in \mathbb{R}^d$ be *concept vectors*. Stack them into the matrix

$$C = [c_1 \ c_2 \ \cdots \ c_n] \in \mathbb{R}^{d \times n}.$$

For any $x \in \mathbb{R}^d$, the vector of concept *scores* (dot products) is

$$C^\top x \in \mathbb{R}^n, \qquad (C^\top x)_j = c_j^\top x.$$

**Erase one concept while preserving the others.** Fix an index $i \in \{1, \ldots, n\}$. Define the current score vector

$$v := C^\top e \in \mathbb{R}^n.$$

We want a modified embedding $r \in \mathbb{R}^d$ such that

$$c_i^\top r = 0, \qquad c_j^\top r = c_j^\top e \text{ for all } j \neq i,$$

and among all such $r$, we want the one with minimal Euclidean change $\|r - e\|_2$.

Define a target score vector $t \in \mathbb{R}^n$ by

$$t_j = \begin{cases} 0 & j = i, \\ v_j & j \neq i. \end{cases}$$

Equivalently, if $e_i \in \mathbb{R}^n$ denotes the $i$-th standard basis vector, then

$$t = v - v_i e_i.$$

The constraints above can be written compactly as

$$C^\top r = t.$$

**Constrained optimization form.** We therefore consider the projection problem

$$\min_{r \in \mathbb{R}^d} \frac{1}{2} \|r - e\|_2^2 \quad \text{s.t.} \quad C^\top r = t. \tag{2}$$

Geometrically, this is the Euclidean projection of $e$ onto the affine subspace $\{r \in \mathbb{R}^d : C^\top r = t\}$.

### C.2 Lagrangian and KKT conditions (full column rank case)

Introduce a Lagrange multiplier $\lambda \in \mathbb{R}^n$ and form the Lagrangian

$$\mathcal{L}(r, \lambda) = \frac{1}{2} \|r - e\|_2^2 + \lambda^\top (C^\top r - t).$$

**Stationarity in $r$** Differentiate with respect to $r$:

$$\nabla_r \mathcal{L}(r, \lambda) = (r - e) + C\lambda.$$

Setting this to zero yields

$$r = e - C\lambda. \tag{3}$$

**Primal feasibility** The constraint is

$$C^\top r = t. \tag{4}$$

Substituting (3) into (4) gives

$$C^\top(e - C\lambda) = t \quad \Longleftrightarrow \quad C^\top e - (C^\top C)\lambda = t.$$

Define the Gram matrix

$$G := C^\top C \in \mathbb{R}^{n \times n}.$$

Then we obtain the linear system

$$G\lambda = C^\top e - t. \tag{5}$$

**If $G$ is invertible.** If the columns of $C$ are linearly independent, then $G$ is invertible and

$$\lambda = G^{-1}(C^\top e - t).$$

Plugging into (3) yields the closed form

$$r = e - C\,G^{-1}(C^\top e - t). \tag{6}$$

### C.3 Rank-deficient case and the pseudoinverse

If the concept vectors are linearly dependent (or nearly so), $G = C^\top C$ may be singular, so (5) can have multiple solutions $\lambda$. However, the primal problem (2) has a *unique* minimizer $r$: the objective $\frac{1}{2} \|r - e\|_2^2$ is strictly convex in $r$ and the feasible set is affine.

A standard way to express the solution is via the Moore–Penrose pseudoinverse $G^+$, which returns the minimum-norm solution of (5):

$$\lambda = G^+(C^\top e - t).$$

Substituting into (3) gives the general closed form

$$r = e - C\,G^+(C^\top e - t), \qquad G = C^\top C. \tag{7}$$

### C.4 Specialization: erasing a single concept $c_i$

Recall $v = C^\top e$ and $t = v - v_i e_i$. Then

$$C^\top e - t = v - (v - v_i e_i) = v_i e_i = (c_i^\top e)\, e_i.$$

Plugging into (7) yields

$$r = e - (c_i^\top e)\, C\,G^+ e_i. \tag{8}$$

This shows the correction is a linear combination of the concept vectors (the columns of $C$), chosen so that only the $i$-th concept score is altered (set to zero) while the others are preserved.

**Orthonormal concept vectors (sanity check).** If the concept vectors are orthonormal, then $G = I$ and $G^+ = I$, so

$$C\,G^+ e_i = C e_i = c_i \implies r = e - (c_i^\top e)\, c_i,$$

which is the usual subtraction of the projection onto $c_i$.

### C.5 Erasing an arbitrary subset of concepts: mask/selector form

Let $S \subseteq \{1, \ldots, n\}$ be a subset of concept indices to erase, while preserving the others. Define the diagonal selector (mask) matrix $P_S \in \mathbb{R}^{n \times n}$ by

$$(P_S)_{jj} = \begin{cases} 1 & j \in S, \\ 0 & j \notin S. \end{cases}$$

Let $v = C^\top e$. The target score vector $t$ that keeps the non-erased scores and zeroes the erased ones is

$$t = (I - P_S)v.$$

Then

$$C^\top e - t = v - (I - P_S)v = P_S v = P_S(C^\top e).$$

Substituting into (7) gives the one-shot multi-erase formula

$$r = e - C\,G^+ P_S\,(C^\top e), \qquad G = C^\top C. \tag{9}$$

This is the unique minimum-change embedding that enforces $c_j^\top r = 0$ for $j \in S$ and preserves $c_j^\top r = c_j^\top e$ for $j \notin S$.

# D  More Details on CLIP-based Representation Surrogates

Let $M^{CLIP}$ refer to the CLIP model, $M_{img}^{CLIP} : Img \to \mathbb{Z}$ refer to the image encoder of CLIP and $M_{txt}^{CLIP} :$ $Txt \to \mathbb{Z}$ refer to its text encoder. For a concept $c \in \mathcal{C}$, its corresponding concept vector $\vec{c}$ is is a vector in $\mathbb{Z}$ given by,

$$\vec{c} := \frac{\Sigma_{t \in Captions(c)} M_{txt}^{CLIP}(t)}{|Captions(c)|}$$

where $Captions(c)$ refers to a set of textual captions referring to concept $c$.

We use the following set of caption templates to generate captions that refer to concepts or classes. The resulting captions are then passed through CLIP's text encoder to generate the text embedding. The actual captions are generated by replacing {} with a class or concept name.

```
caption_templates = [                        'a sketch of a {}.',
    'a bad photo of a {}.',                  'a doodle of the {}.',
    'a photo of many {}.',                   'a low resolution photo of a {}.',
    'a photo of the hard to see {}.',        'a photo of the clean {}.',
    'a low resolution photo of the {}.',     'a photo of a large {}.',
    'a rendering of a {}.',                  'a photo of a nice {}.',
    'a bad photo of the {}.',                'a photo of a weird {}.',
    'a cropped photo of the {}.',            'a blurry photo of a {}.',
    'a photo of a hard to see {}.',          'a cartoon {}.',
    'a bright photo of a {}.',               'art of a {}.',
    'a photo of a clean {}.',                'a sketch of the {}.',
    'a photo of a dirty {}.',                'a pixelated photo of a {}.',
    'a dark photo of the {}.',               'a jpeg corrupted photo of the {}.',
    'a drawing of a {}.',                    'a good photo of a {}.',
    'a photo of my {}.',                     'a photo of the nice {}.',
    'a photo of the cool {}.',               'a photo of the small {}.',
    'a close-up photo of a {}.',             'a photo of the weird {}.',
    'a black and white photo of the {}.',    'the cartoon {}.',
    'a painting of the {}.',                 'art of the {}.',
    'a painting of a {}.',                   'a drawing of the {}.',
    'a pixelated photo of the {}.',          'a photo of the large {}.',
    'a bright photo of the {}.',             'a black and white photo of a {}.',
    'a cropped photo of a {}.',              'a dark photo of a {}.',
    'a photo of the dirty {}.',              'a photo of a cool {}.',
    'a jpeg corrupted photo of a {}.',       'a photo of a small {}.',
    'a blurry photo of the {}.',             'a photo containing a {}.',
    'a photo of the {}.',                    'a photo containing the {}.',
    'a good photo of the {}.',               'a photo with a {}.',
    'a rendering of the {}.',                'a photo with the {}.',
    'a {} in an image.',                     'a photo containing a {} object.',
    'a photo of one {}.',                    'a photo containing the {} object.',
    'a doodle of a {}.',                     'a photo with a {} object.',
    'a close-up photo of the {}.',           'a photo with the {} object.',
    'a photo of a {}.',                      'a photo of a {} object.',
    'the {} in an image.',                   'a photo of the {} object.']
```

Following  Bhalla et al. (2024), where the authors empirically found that CLIP image and text embeddings exist on two cones of the embedding space, they suggest applying a mean-centering procedure followed by a normalization operation to both concept vectors and image embeddings, using a concept mean and an image mean, respectively, computed over a entire vocabulary and all images of a dataset of interest; in our case we

use the vocabulary conformed by the 13K most frequent words in MSCOCO captions and the RIVAL10 and EuroSAT datasets consisting of 20K and 27K images each. We apply this procedure to class vectors as well.

**Definition D.1** (Mean-Centering in CLIP Space). For any concept $c \in \mathcal{C}$, class $k \in \mathcal{K}$, or image $e \in Img$, their corresponding mean-centered vectors in CLIP's embedding space are vectors in $\mathbb{Z}$ given by,

$$\xi_{txt}(x) := \sigma \left( \frac{\Sigma_{t \in Captions(x)} M_{txt}^{CLIP}(t)}{|Captions(x)|} - \mu_{txt} \right), x \in \mathcal{C} \cup \mathcal{K} \quad \text{and} \quad \mu_{txt} = \frac{\Sigma_{t' \in \mathcal{C}_{\text{MSCOCO}}} M_{txt}^{CLIP}(t')}{|\mathcal{C}_{\text{MSCOCO}}|}$$

$$\xi_{img}(e) := \sigma \left( M_{img}^{CLIP}(e) - \mu_{img} \right), e \in Img \quad \text{and} \quad \mu_{img} = \frac{\Sigma_{e' \in Img} M_{img}^{CLIP}(e')}{|Img|}$$

where $\sigma(x) = x/||x||_2$ refers to the normalization operation.

### D.1 Sanity Checks for Linear Maps Between Embedding Spaces

We validate the linear mapping discussed in Section 2.2.1 between vision model and CLIP embedding spaces to ensure the transformation preserves concept-related information. Related work Mangal et al. (2024); Merullo et al. (2023) uses linear maps to transform embeddings from one model to match embeddings of another model for the same inputs. We go one step further by mapping these vectors back to the original vision model space. This inverse mapping ensures that edits in the CLIP space, such as concept erasure, remain effective after returning to the original model. We measure this consistency using Concept Activation Vectors (CAVs) Kim et al. (2018). Each CAV is obtained by training a Ridge linear model[3] to distinguish embeddings containing a specific concept from those that do not. We first formally define Inverse Maps and CAV-based Concept Strength to provide the foundation for our three sanity checks.

**Definition D.2** (Inverse Map). Given a set of images $Img$, a vision model $M^{vis}$, and a linear map from vision model $M^{vis}$ to $M^{CLIP}$ embedding spaces $map_{M^{vis} \mapsto M^{CLIP}}(x) = W \cdot M^{enc}(x) + d \approx M_{enc}^{CLIP}(x) : x \in Img$ denoted simply as $map(x)$, we say that its corresponding inverse map, denoted as $map^{-1}(x)$, is a linear map from $M^{CLIP}$ to vision model $M^{vis}$ embedding spaces such that $\forall_{x \in Img} map^{-1}(map(x)) = x$.

We explored two ways to obtain an Inverse Map. The first method relies on finding the matrix $W_{inv}$ that solves the expression $W_{inv} \cdot W = I$. The best solution is the *Moore-Penrose pseudoinverse*, computed via the `PyTorch` function `torch.linalg.pinv(W)`. However, the `PyTorch` documentation[4] recommends instead using their least squares solver `torch.linalg.lstsq(W, I)` to obtain the pseudoinverse $W_{inv}$ because it is faster and more numerically stable than computing the solution directly. The inverse mapping is then:

$$map_{lstsq}^{-1} = W_{inv} \cdot (map(x) - d) \qquad \text{s.t.} \qquad \min_{W_{inv}} \|W_{inv} \cdot W - I\|_F$$

The second method involves learning a new mapping by using the original training data in reverse.

$$map_{learnt}^{-1} = \hat{W} \cdot map(x) + \hat{d} \qquad \text{s.t.} \qquad \hat{W}, \hat{d} = \arg\min_{\hat{W}, \hat{d}} \frac{1}{|D_{train}|} \sum_{x \in D_{train}} \|\hat{W} \cdot M_{img}^{CLIP}(x) + \hat{d} - M^{enc}(x)\|_2^2$$

**Definition D.3** (CAV-based Concept Strength). Given a concept $c \in \mathcal{C}$, a set of images tagged by a human analyst as containing concept $c$ denoted *positive set* $P_c$, a set of images that do not (e.g., a set of random images) denoted *negative set* $N$, a vision model encoder $M^{enc}$, a linear regressor $f$ trained to distinguish between the embeddings of the two sets $\{f(M^{enc}(x)) = 1 : x \in P_c\}$ and $\{f(M^{enc}(x)) = 0 : x \in N\}$, and an image $\psi$, we say that the CAV-based Strength of concept $c$ in image $\psi$ is $CAV_c(\psi) = f(M^{enc}(\psi))$.

While both methods provide Inverse Maps that satisfy $map^{-1}(map(x)) \approx x$, we decided to use $map_{learnt}^{-1}$ because it generates more stable results when concept erasure is applied in CLIP space. After all, erasing a concept in an embedding forces that embedding to be slightly out of its original distribution. We observe in Table 4 that $map_{lstsq}^{-1}$ do not handle such slightly out-of-distribution embeddings appropriately, and report

---

[3]See https://scikit-learn.org/stable/modules/generated/sklearn.linear__model.Ridge.html
[4]See https://docs.pytorch.org/docs/stable/generated/torch.linalg.pinv.html

Table 4: Comparison of methods for obtaining inverse linear maps. The subset $e \subseteq Img$ has been manually selected to contain images where $c_1$ is present and $c_2$ is not.

| Inverse Map | Concept | $CAV_{c_i}(e)$ | $CAV_{c_i}(map^{-1}(erase(map(e), c_1)))$ | $CAV_{c_i}(map^{-1}(erase(map(e), c_2)))$ |
|---|---|---|---|---|
| $map^{-1}_{learnt}$ | $c_1$=Wheels | 0.9186 | 0.7857 | 0.8445 |
| | $c_2$=Ears | -0.0146 | 0.0471 | 0.0296 |
| $map^{-1}_{lstsq}$ | $c_1$=Wheels | 0.9250 | 10.6065 | -0.1577 |
| | $c_2$=Ears | 0.0753 | -4.5047 | -3.5796 |

Table 5: Sanity checks for linear mapping between embedding spaces. In each row, the subset $e \subseteq Img$ has been manually selected to contain images where $c_1$ is present and $c_2$ is not.

| Backbone | Concept | $CAV_{c_i}(e)$ | $CAV_{c_i}(map^{-1}(map(e)))$ | $CAV_{c_i}(map^{-1}(erase(map(e), c_1)))$ | $CAV_{c_i}(map^{-1}(erase(map(e), c_2)))$ |
|---|---|---|---|---|---|
| zsCLIP | $c_1$=Wings | 0.1076 | 0.1036 | -0.3502 | 0.0632 |
| | $c_2$=Horns | 0.7588 | 0.5080 | 0.5155 | 0.3196 |
| ResNet18 | $c_1$=Wings | 0.1110 | 0.1837 | -0.2726 | 0.1355 |
| | $c_2$=Horns | 0.7661 | 0.6457 | 0.6101 | 0.4179 |
| VGG19 | $c_1$=Wings | 0.0029 | 0.3345 | 0.1870 | 0.3388 |
| | $c_2$=Horns | 0.9873 | 0.4368 | 0.4347 | 0.2805 |

nonsensical CAV-based Concept Strengths such as increasing the strength of a concept that has been erased, or nullifying the strength of concept $c_1$ when concept $c_2$ has been erased.

**Sanity Checks.** First, we check whether concepts retain similar CAV-based strength after mapping to CLIP and then back to the vision model. Second, we check that a concept erased in the CLIP space remains erased after mapping the embedding back to the vision model. Third, we check that erasing concept $c_i$ in CLIP space does not significantly change the strength of an unrelated concept $c_j$. Formally expressed as:

$$Check_1(x, c) := \frac{|CAV_c(x) - CAV_c(map^{-1}(map(x)))|}{|CAV_c(x)|} \leq \beta \tag{10}$$

$$Check_2(x, c) := CAV_c(map^{-1}(map(x))) > CAV_c(map^{-1}(erase(map(x), c))) \tag{11}$$

$$Check_3(x, c_A, c_B) := \frac{|CAV_{c_A}(x) - CAV_{c_A}(map^{-1}(erase(map(x), c_B)))|}{|CAV_{c_A}(x)|} \leq \gamma \tag{12}$$

Table 5 show the resulting CAV-based Concept Strengths of different models and their corresponding linear maps. All sanity checks hold true, reporting $\beta \in [0.037, 0.65]$ and $\gamma \in [0.0048, 0.390]$ on zsCLIP and ResNet18; VGG19 shows significantly lower performance compared to them, as indicated by larger $\beta$ and $\gamma$ values. This result likely reflects the challenge of mapping between VGG's 4096-dimensional penultimate layer and the 512-dimensional embedding space of CLIP. Related work Koh et al. (2020) suggest that this mapping could improve by training a small multi-layer perceptron, but we decided to keep the mapping linear in this study.

## E  Concept Vocabulary

Identifying which and how many concepts an ideal vocabulary should contain remains an open problem, for which various approaches have been utilized in the literature: leveraging human-labeled datasets Mangal et al. (2024), semantic pruning from a base dictionary Bhalla et al. (2024); Oikarinen et al. (2023), interpreting features from learned dictionaries Cunningham et al. (2023), asking domain experts (dermatologists, radiologists, and birders) for guidelines used in the field (visual melanoma detection, osteoarthritis severity measurement from knee x-rays, and identify bird species) Dreyer et al. (2025); Koh et al. (2020), and relying on Language Models' output Li et al. (2025); Dunlap et al. (2024).

In this work, we build upon the vocabulary design from Bhalla et al. (2024) which starts with a task-agnostic base vocabulary and then prunes it. This ensures broad applicability of the vocabulary across diverse downstream prediction tasks. The base vocabulary comprises of approximately 13K words that appear most frequently in MSCOCO captions Lin et al. (2015). To ensure all words are meaningful, we intersect this set with CoreWordNet Khodak et al. (2017); Clark et al. (2008), a lexicon where each word possesses a well-defined meaning. This process filters out common function words (e.g., 'a', 'the', 'in') that lack conceptual weight. Finally, to minimize redundancy, we use CLIP text embeddings to compute cosine similarity between all pairs of concepts and, for each pair where similarity exceeds 0.9, we prune the less frequent concept. We do the same for concepts too similar to the target classes. Others Bhalla et al. (2024); Oikarinen et al. (2023) have proposed similar concept filtering processes to prevent duplicate concepts. After pruning, we end up with an *intermediate* vocabulary of 1500 concepts.

**Determining Optimal Vocabulary Size.** There exists a size trade-off between a vocabulary comprehensive enough to explain most instances of a behavior of interest and one small enough to ensure that enumeration of explanations remains computable within a given time-out. We define the following test to evaluate if a vocabulary contains enough concepts.

**Definition E.1** (Vocabulary Size Test)**.** Given a concept vocabulary $\mathcal{C}$, a set of images $Img$, and a model $M^{vis}$, we say that the vocabulary is $\alpha$-suitably sized if,

$$\frac{|\{v \in Img \mid M^{vis}(v) = k \wedge M^{head}(erase(M^{enc}(v), \mathbf{0})) \neq k\}|}{|Img|} \geq \alpha$$

where $\mathbf{0}$ is the zero vector of length $|\mathcal{C}|$ indicating that all the concepts in the vocabulary should be erased. The intuition is that even if erasing all concepts does not cause the vision model to change its prediction on an $\alpha$-sized fragment of the dataset, then the concept vocabulary is not expressive enough. Note that if erasing all the concepts in an image's embedding causes the prediction to change, then it indicates that there is at least one potential contrastive and abductive explanation for that image—the one that includes all concepts.

Given this test, we proceed to further prune the intermediate vocabulary of 1500 concepts such that it is as small as possible while ensuring high $\alpha$ values (where $\alpha$ is specified by the analyst). To do so, we first order the concepts in the intermediate vocabulary and then consider each of the vocabularies comprising of the first $n$ concepts for all $n \leq 1500$. To order the concepts, this study leverages the empirical observation that ordering concepts by their average absolute activation strength across a dataset, with respect to a specific model, results in higher $\alpha$ values for that model than baselines such as frequency-based ordering of concepts, e.g., Bhalla et al. (2024) prioritize concepts based on frequency over MSCOCO captions. Our ordering criterion is in line with the pruning procedure used by Oikarinen et al. (2023), where concepts were filtered-out from their vocabulary if their average top-5 activation strength was below some dataset-specific cut-off value, and Papadimitriou et al. (2025), where they define the *energy of concept i in a given dataset* as its average activation strength across that dataset.

We repeat this process for each of the three erasure algorithms detailed in Subsection 4.1. Note, when testing with SpLiCE-based erasure, we have to account for the fact that the embedding can change even if no concepts are erased since SpLiCE first computes a sparse decomposition of the original embedding and then computes a new embedding from this sparse decomposition. We discard any image for which SpLiCE's concept decomposition leads to a prediction different from that of the unmodified embedding, i.e., such images are not included in the numerator of the expression in Definition E.1.

Figure 11 shows the results of the test for different sizes of the vocabulary. Consider Figure 11b, it shows that for zero-shot CLIP, a vocabulary of 50 concepts leads to $\alpha = 0.875$ using LEACE, but results in $\alpha < 0.5$ for Ortho and SpLiCE. On the other hand, although selecting 250 or more concepts leads to the highest $\alpha$ values, the computation time for enumerating all explanations given such large vocabularies can be prohibitive. A sweet spot occurs at 150 concepts, where all three erasure procedures reach an $\alpha$ value of at least 0.75. In contrast, using the baseline criterion (dotted line), 150 concepts result in an $\alpha$ below 0.375 for two erasure methods. Reaching the $\alpha = 0.75$ mark with the baseline requires 650 concepts. Our approach uses the

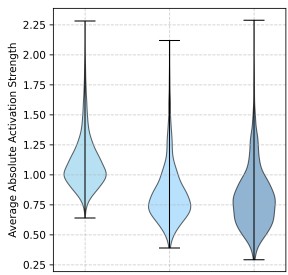
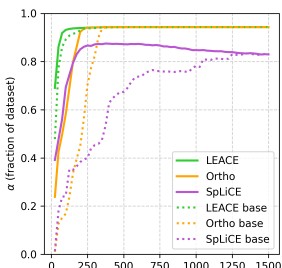
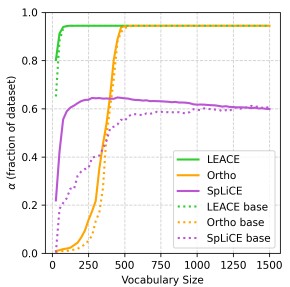
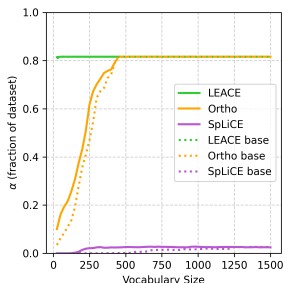

(a) Intermediate vocabulary Average Absolute Activation Strength distribution

(b) Explainable dataset fraction $\alpha$ vs. Vocabulary Size for zero-shot CLIP

(c) Explainable dataset fraction $\alpha$ vs. Vocabulary Size for ResNet18

(d) Explainable dataset fraction $\alpha$ vs. Vocabulary Size for VGG19

Figure 11: Selecting concepts based on (a) average absolute activation strength leads to optimal $\alpha$ values across models (b–d), including (b) zero-shot CLIP, (c) ResNet18, and (d) VGG19 on the dataset *RIVAL10* (higher $\alpha$ is better). Baseline (dotted line in b–d): frequency-based criteria for ordering concepts.

vocabulary sizes detailed in Table 1 for every model and erasure method combination. These vocabularies are used for all experiments described in the following sections.

**Customized Vocabularies.** Beyond the specific vocabularies used in our evaluation, a key strength of our approach is its flexibility, as it is not restricted to a specific vocabulary. Concepts can be selected to suit specific needs, such as limiting the set to visual concepts for better visual grounding or incorporating domain-specific concepts to improve explanatory accuracy in specialized tasks, such as melanoma classification Dreyer et al. (2025). This adaptability allows for the personalization of explanations to match the background knowledge of different audiences, e.g., by selecting accessible concepts in educational settings. Furthermore, the ability to adjust the vocabulary aligns with the definitions of interactive and actionable explanations found in the literature Sokol & Flach (2020), as it enables users to refine granularity to suit their needs and provides concrete actions toward a desired outcome.

# F    Maximum Coverage@K Algorithm

We formulate the selection of explanations in Section 5.2.2 as an instance of the Maximum Coverage Problem. Given a collection of subsets $\{b_{x_1}, \ldots, b_{x_n}\}$ representing images in $B_{M^{vis}}$ and an integer $K$, the goal is to select a subset of at most $K$ explanations that maximizes the total number of unique images covered. Since this problem is NP-hard, we employ a greedy strategy that iteratively selects the next explanation with the highest marginal gain in image coverage. This approach, adapted in Algorithm 5 to our needs, is one of the fastest yet simple to implement polynomial time approximations Hochbaum (1997).

# G    Additional Experimental Results

The main text employs three reductions when reporting results: omitting model $M_1$, including only a subset of the explored behaviors, and collapsing results by enumeration algorithm. This compact way of displaying information aids readers' understanding at the expense of not reporting data in a fine-grained way.

This appendix provides the fine-grained results omitted from the main text. There are 12 behaviors included, 4 for each vision model backbone: ResNet18, VGG19, and zero-shot CLIP. These 4 behaviors are divided not only into two correct and two incorrect classification behaviors, but also into two behaviors per dataset explored: RIVAL10 and EuroSAT. This arrangement provides one behavior for each combination, so the reader has a small but complete picture of the empirical results. This arrangement applies to Figures 12 to 18; with the exception of Figure 17 and Table 6, which contain just eight behaviors each, complementing the four shown in Figure 10 and Table 2, respectively. For the rest of this section, we define $M_1$ as CLIP used

---

**Algorithm 5:** Maximum Coverage@K (Greedy Selection)

---

**Input:** Behavior $B_{M^{vis}}$, Explanation sets $\hat{\mathcal{X}}_A$ and $\hat{\mathcal{X}}_C$, Coverage subsets $\{b_x \mid x \in \hat{\mathcal{X}}_A \cup \hat{\mathcal{X}}_C\}$, Size $K$
**Output:** Pair of subsets $(S_A, S_C)$

**1** $S_A \leftarrow \emptyset$, $U_{covered} \leftarrow \emptyset$;
**2 for** $i = 1$ **to** $K$ **do**
**3**     $x \leftarrow \arg\max_{x \in \hat{\mathcal{X}}_A \setminus S_A} |b_x \setminus U_{covered}|$ ;            // Select $x$ with highest marginal gain
**4**     $S_A \leftarrow S_A \cup \{x\}$;
**5**     $U_{covered} \leftarrow U_{covered} \cup b_x$;
**6** $S_C \leftarrow \emptyset$, $U_{covered} \leftarrow \emptyset$;
**7 for** $i = 1$ **to** $K$ **do**
**8**     $x \leftarrow \arg\max_{x \in \hat{\mathcal{X}}_C \setminus S_C} |b_x \setminus U_{covered}|$ ;            // Select $x$ with highest marginal gain
**9**     $S_C \leftarrow S_C \cup \{x\}$;
**10**    $U_{covered} \leftarrow U_{covered} \cup b_x$;
**11 return** $(S_A, S_C)$;

---

Table 6: Average quantity of explanations found per image and its standard deviation for additional behaviors.

| Behavior | | Ortho | | | SpLiCE | | | LEACE | | |
|---|---|---|---|---|---|---|---|---|---|---|
| | | XpEnum | XpSatEn. | NaiveEn. | XpEnum | XpSatEn. | NaiveEn. | XpEnum | XpSatEn. | NaiveEn. |
| $B_{M_3}$(AnnualCrop) | ConAXps | 11.9±6.8 | 173.9±18.4 | 57.0±30.0 | 12.3±2.9 | 18.9±2.5 | 1.7±18.4 | - | - | - |
| | ConCXps | 6.2±3.6 | 67.2±14.5 | 267.7±16.5 | 4.3±6.3 | 8.5±4.4 | 1.6±19.5 | 10.0±6.1 | 77.7±9.7 | 723.5±12.0 |
| $B_{M_3}$(River,AnnualCrop) | ConAXps | 170.0±0.2 | 1480.3±0.8 | 21.1±1.5 | - | - | - | - | - | - |
| | ConCXps | 20.6±0.2 | 254.2±0.9 | 90.6±1.3 | - | - | - | 10.0±2.4 | 45.3±5.2 | 1349.3±6.2 |
| $B_{M_4}$(Deer,Car) | ConAXps | 16.6±0.2 | 25.4±0.5 | 99.0±0.5 | - | - | - | - | - | - |
| | ConCXps | 6.6±0.2 | 13.2±0.9 | 251.0±0.5 | - | - | - | 10.0±0.1 | 25.6±0.8 | 344.2±0.5 |
| $B_{M_5}$(AnnualCrop) | ConAXps | 6.6±4.3 | 24.7±8.6 | 223.5±4.7 | - | - | - | - | - | - |
| | ConCXps | 3.8±1.3 | 24.5±6.2 | 260.0±3.4 | - | - | - | 9.9±4.5 | 34.7±9.3 | 1354.2±7.5 |
| $B_{M_1}$(Deer) | ConAXps | 60.4±5.4 | 1432.4±14.4 | 123.2±23.3 | 29.1±0.6 | 65.2±1.2 | 0.0±4.6 | - | - | - |
| | ConCXps | 8.7±0.2 | 1096.2±2.9 | 3.6±1.3 | 20.2±6.1 | 66.8±7.6 | 15.2±24.2 | 10.0±0.6 | - | 206.2±23.5 |
| $B_{M_1}$(Equine,Car) | ConAXps | 38.1±0.3 | 70.7±0.7 | 122.4±0.7 | 0.2±0.0 | 0.4±0.0 | 1.1±1.0 | - | - | - |
| | ConCXps | 8.3±0.0 | 22.4±0.5 | 95.8±0.5 | 0.2±0.0 | 0.3±0.0 | 101.2±0.3 | 10.0±0.3 | 27.7±1.1 | 300.0±0.7 |
| $B_{M_1}$(Frog) | ConAXps | 84.5±5.8 | 1711.6±15.3 | 70.7±20.1 | 15.4±0.4 | 27.4±0.5 | 0.0±0.0 | - | - | - |
| | ConCXps | 10.8±0.1 | 1910.5±2.2 | 0.1±2.1 | 14.0±12.5 | 30.2±11.9 | 30.0±52.4 | 10.0±0.8 | - | 75.0±33.9 |
| $B_{M_1}$(PermaCrop,Highway) | ConAXps | 24.1±6.2 | 139.8±8.3 | 16.3±16.3 | 15.7±1.7 | 28.9±2.0 | 1.0±9.0 | - | - | - |
| | ConCXps | 9.4±2.2 | 276.3±2.9 | 19.8±5.3 | 11.0±2.9 | 24.9±3.0 | 7.1±5.6 | - | - | 83.0±8.3 |

as a zero-shot classifier. In this *ad-hoc* vision model, the vision encoder is CLIP itself, and the vision head is an argmax of the cosine similarity between an embedding and the class vectors.

**Definition G.1** (Zero-shot Classification with CLIP). We define the model $M^{CLIP}$ as a function of type $Img \rightarrow \mathcal{K}$ which can be decomposed into $M^{CLIP} = M^{CLIP}_{head} \circ M^{CLIP}_{enc}$ where $M^{CLIP}_{enc} : Img \rightarrow \mathbb{Z}$ and $M^{CLIP}_{head} : \mathbb{Z} \rightarrow \mathcal{K}$ where $\mathbb{Z}$ is the representation (or embedding) space.

$$M^{CLIP}_{enc}(x) = z, x \in Img, z \in \mathbb{Z} \quad \text{and} \quad M^{CLIP}_{head} = \arg\max_i cos(z, \vec{k_i}), k_i \in \mathcal{K}$$

where $\vec{k_i}$ is the i-th class vector (obtained the same way as concept vectors, Appendix D) of the class set $\mathcal{K}$.

The rest of this appendix is divided into 8 subsections: i-iv) supplementary results for RQs 1-4, v) an in-depth discussion about the efficiency of computing ConXps, provided as an extension to our RQ5 in Section 5.2.5, vi) "Relative Cumulative Frequency at Length $K$", a complementary metric not described in the main text, vii) "Signed Concept-based Explanations" examples, and viii) "Qualitative Analysis", with further discussion and use cases about how ConXps can assist human analysts in evaluating vision models' behaviors.

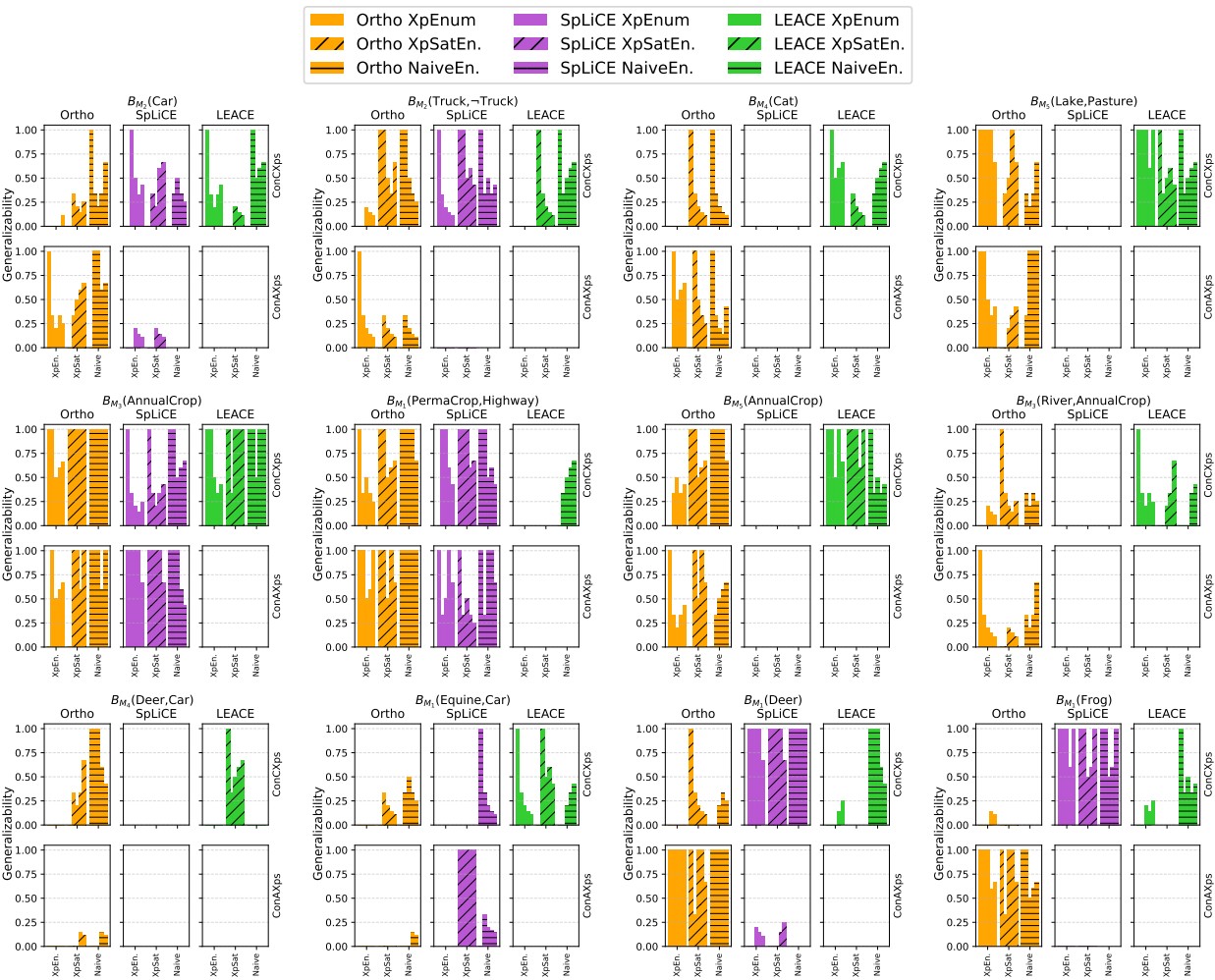

Figure 12: Supplementary experimental results for RQ1. (Metric: Generalizability@K)

## G.1 RQ1: Generalizability

Supplementary results for the Generalizability@K metric across all explored behaviors are reported in Figure 12. Each behavior is represented as a $2 \times 3$ matrix of subplots, where the columns correspond to erasure algorithms, and the rows correspond to ConCXps and ConAXps, respectively. Inside each subplot, results for all three explanation enumeration algorithms are presented, when available. The data consists of five columns, each one with a Generalizability score at $K \in [1, 5]$; higher is better, following the logic that a high value of Intersection-over-Union (IoU) tells that those ConXps are applicable across different samples of the same behavior, therefore suggesting that they capture the essence of why the model exhibits a specific behavior. Note that some individual bars appear to be missing; this is the case when their IoU value is equal to zero. See Section 5.2 for a discussion about why some whole series are missing, e.g., LEACE ConAXps.

## G.2 RQ2: Coverage

Supplementary results for the MaximumCoverage@K and Individual Coverage metrics across all explored behaviors are reported in Figures 13 and 14, respectively. In both figures, each behavior is represented as a line plot containing up to 18 series. Each series reports an individual combination of three factors: i) erasure algorithm, shown using different colors, ii) explanation enumerator, differentiated via line patterns (straight, dashed, and dotted), and iii) ConAXps and ConCXps shown with and without markers, respectively.

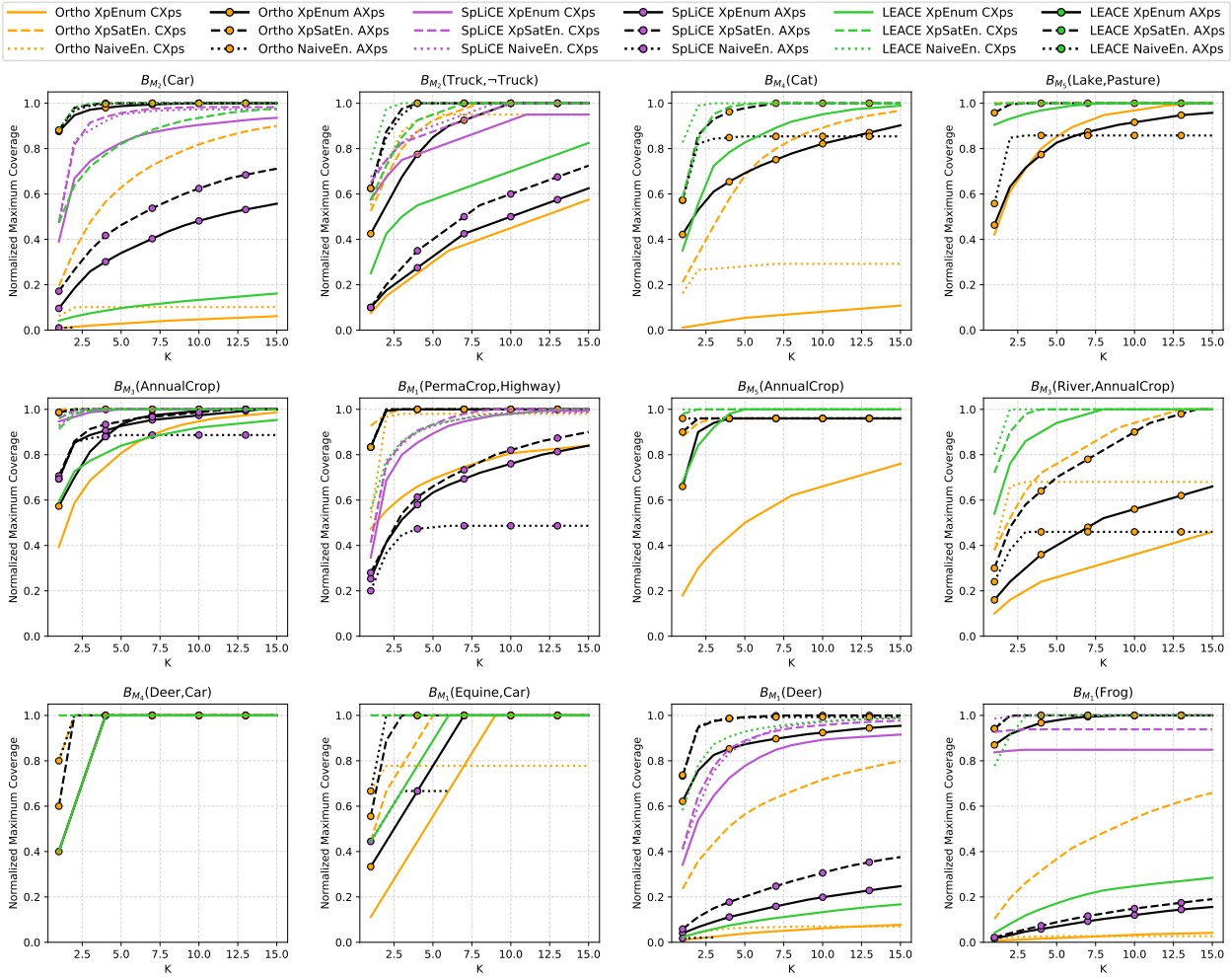

Figure 13: Supplementary experimental results for RQ2. (Metric: MaximumCoverage@K)

Figure 13 shows that LEACE × NaiveEnum × ConCXps (green dotted line with no markers) and Ortho × XpSatEnum × ConAXps (black dashed line with orange markers) consistently report high MaximumCoverage@K values, while Ortho × XpEnum × ConCXps (orange line with no markers) and SpLiCE × XpEnum × ConAXps (black line with purple markers) underperform in the majority of the cases. In addition to that, Figure 14 shows Individual Coverage values that are consistent with the previous observation.

## G.3 RQ3: Parsimony

Supplementary results for the comparison of ConXps' length vs. Individual Coverage across all explored behaviors are reported in Figure 15. Each behavior is represented as a $3 \times 1$ matrix of subplots, where each rows correspond to an explanation enumerator. Inside each subplot, results for all three erasure algorithms × ConXps are presented, when available. See Section 5.2 for a discussion about why the combination LEACE × ConAXps was not included. The data consists of three violin plots, each for a subset of ConXps, depending on their length $L$, defined as the number of concepts included in them, e.g., let $\mathcal{X} = \{con_1, \ldots, con_n\}$ then we say its length is $|\mathcal{X}| = n$. Each violin plot's width at a given y-axis value represents the proportion of explanations reporting that value of individual coverage. The better combination is low $L$ and high individual coverage, following Occam's razor, which favors shorter explanations over longer ones. Note that violin plots can be seen as 'box plots' with variable width alongside the y-axis; also, note that outliers are omitted.

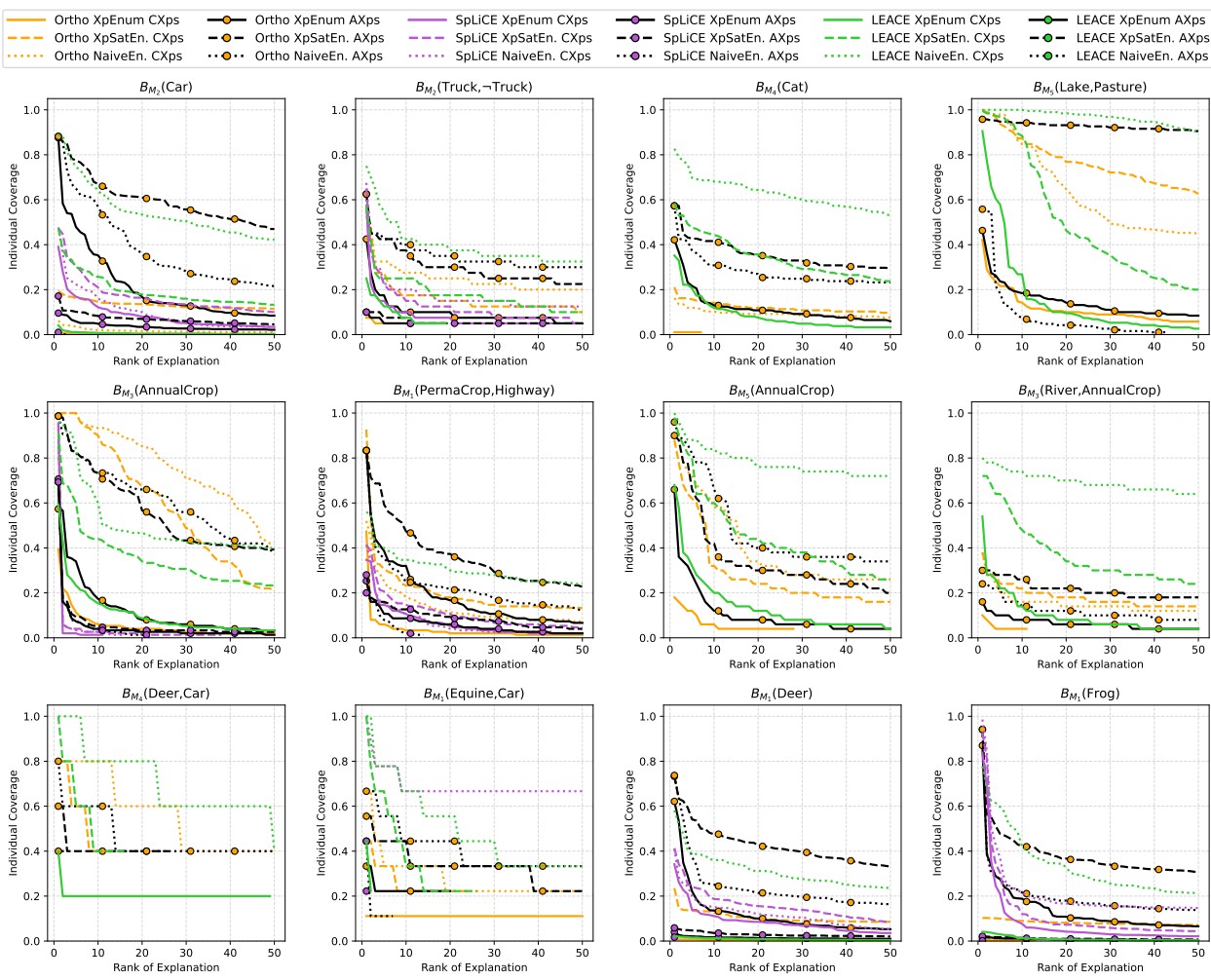

Figure 14: Supplementary experimental results for RQ2. (Metric: Individual Coverage)

### G.4 RQ4: Plausibility

Supplementary results for the Plausibility Ratio metric across all explored behaviors are reported in Figure 16. Each behavior is represented as a $2 \times 3$ matrix of subplots, where the columns correspond to erasure algorithms, and the rows correspond to ConCXps and ConAXps, respectively. Inside each subplot, results for all three explanation enumeration algorithms are presented, when available. The data consists of three columns: fully plausible, partially plausible, and fully implausible explanations, following Section 5.2.4. On average, SpLiCE-based explanations tend to report a higher ratio of fully plausible explanations, although we leave a more comprehensive plausibility analysis as future work. Note that the sum of all three columns equals 100%.

### G.5 RQ5: Efficiency

We compare the time required per combination of an erasure algorithm with an explanation enumeration algorithm to compute explanations per image. We refer to each such combination as a *configuration*.

The three erasure algorithms exhibit varying degrees of reusability of computation which directly impacts their computation times. Ortho utilizes a series of matrix operations that can be pre-calculated and reused, resulting in a short time for execution. Similarly, while SpLiCE initially requires solving a numerical optimization for each image embedding, the resulting concept strengths can be reused to make subsequent

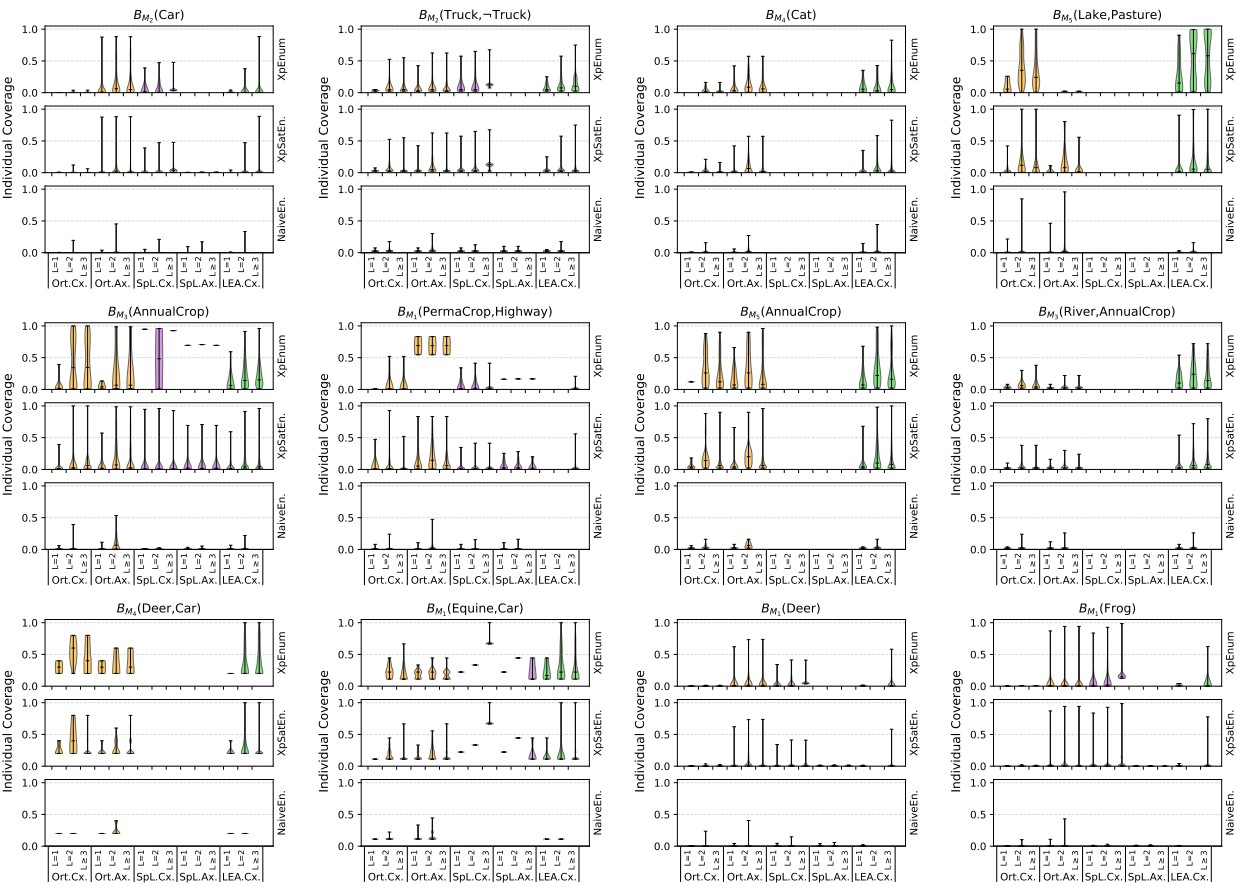

Figure 15: Supplementary experimental results for RQ3. (Metric: |Xp| vs. IndCov)

erasures efficient. In contrast, LEACE uses training data to compute a covariance matrix and calculates an affine transformation for each specific erasure. As a result, while repeating the same erasure is fast, erasing different sets of concepts requires new calculations, leading to longer computation times. A practical short-hand is to store the LEACE intermediate results in memory; however, this is only feasible when repeating identical erasures, such as in cases involving the NaiveEnum algorithm discussed below.

The three explanation enumeration algorithms explore the space of explanations in different manners which impacts their computation times. The computation time for NaiveEnum is primarily determined by the search depth $K$, followed by the vocabulary size and the number of images. By fixing the search depth at $K = 2$, we maintain tractability while focusing on the most effective explanations, which are those with the highest individual coverage as empirically shown in Section 5.2.3. For behaviors consisting of 700 images each and vocabularies between 150 and 300 concepts, both Ortho and SpLiCE show similar computation times, ranging from 1h30m to 2h. In contrast, using LEACE is only feasible if the pre-computed intermediate results are stored for all searched explanations. In these cases, the first image requires a significant portion of the total time, but subsequent images are processed in a shorter time because no further training is involved. While this approach scales well with the number of images, the vocabulary size remains a critical constraint; exploring 150 concepts takes nearly 2h, whereas 500 concepts require about 10h.

In the case of XpEnum (detailed in Appendix B), when used in conjunction with Ortho or SpLiCE, this algorithm is the fastest configuration for generating explanations, reporting computation times between 5 and 10 minutes for behaviors of 700 images each. Empirically, the impact of vocabulary size and the number of images on the computation time is roughly equivalent, which contrasts with NaiveEnum, where vocabulary size was much more impactful than the number of images. The unpredictable way XpEnum explores the search space makes it the second slowest configuration when paired with LEACE. We empirically observe

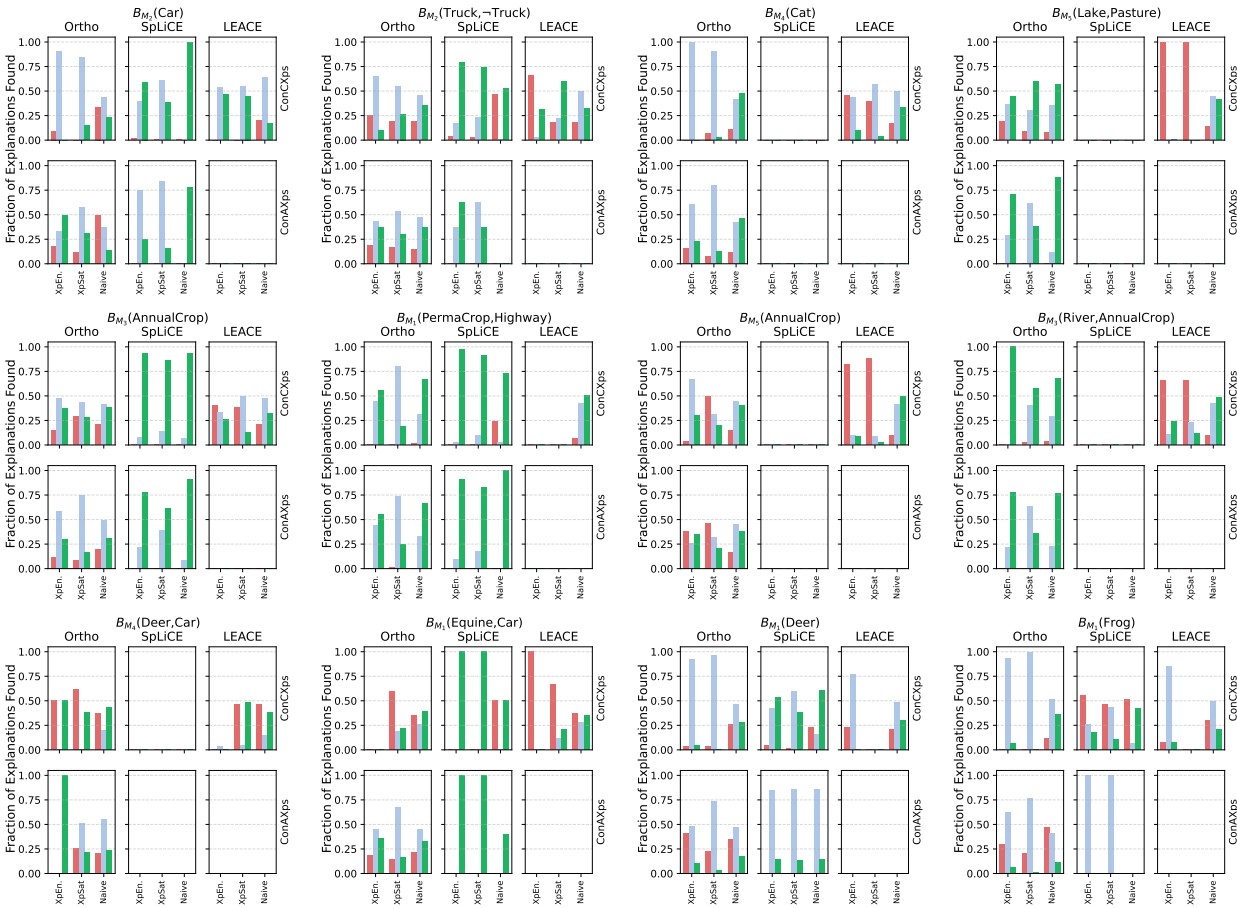

Figure 16: Supplementary experimental results for RQ4. Plots show the fraction of fully plausible (green), partially plausible (light blue), and fully implausible (red) explanations, as defined in Section 5.2.4.

that the algorithm typically finds ConCXps in early iterations, transitions into alternating between ConAXps and ConCXps, and finally focuses exclusively on ConAXps; however, because pre-computed intermediate results are rarely reused during this traversal, computation times frequently exceed our 10h timeout. In these cases, the process must be interrupted, often before any ConAXps are obtained. An in-depth study of this enumeration algorithm to reduce computation time is left as future work.

The computation time for XpSatEnum is directly determined by the number of discovered explanations and the images to be saturated. While vocabulary size plays a role, it is less impactful than these two factors. When paired with Ortho or SpLiCE for behaviors with 700 images each, computation time varies by behavior due to the number of explanations. Ortho averages 5 hours, and SpLiCE averages 2 hours among the behaviors explored, with specific executions spanning from 30 minutes to 8.5 hours. The combination of XpSatEnum and LEACE is the slowest configuration due to scaling bottlenecks that can exceed a 40 GB GPU memory limit. First, the number of explanations to check is in the order of thousands. Second, an explanation that is subset-minimal for one image may only be a weak explanation for others, requiring further pruning to guarantee minimality. This leads to an exponential increase in required checks. In these cases, the lack of reusability across diverse candidates prevents the algorithm from completing in a reasonable time.

Overall, although formal explanation methods often struggle to scale to large models and datasets, our approach offers a reasonable trade-off between scalability and explanation quality. Ortho and SpLiCE paired with XpEnum are the fastest configurations for discovering both ConAXps and ConCXps within a short time. While XpSatEnum is the most computationally expensive due to the checks it performs to

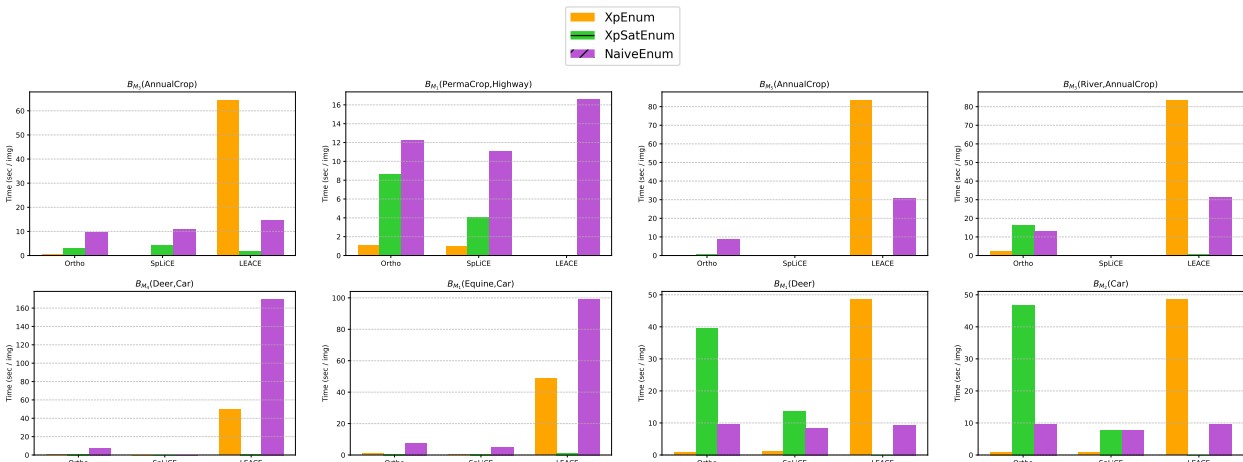

Figure 17: Compute time per image, measured in seconds, for additional behaviors.

guarantee minimality, it offers the advantage of higher maximum coverage than XpEnum by construction. In contrast, LEACE is fastest when paired with NaiveEnum's predictable search pattern for finding short ConCXps, as this configuration leverages precomputed intermediate results.

## G.6   Relative Cumulative Individual Coverage at Length K

We define a complementary metric to quantify the conciseness of ConXps, motivated by the intuition that shorter explanations are more accessible to human analysts. Prior work  Ramaswamy et al. argues that the number of concepts contained in a single explanation should be limited by a strict upper bound of 32 to align with human preferences. This study also notes that the largest percentage of participants prefer fewer than 16 concepts, and the highest accuracy was achieved with a maximum of only 8 concepts per explanation.

To evaluate how our results match these human preferences, we use the Relative Cumulative Individual Coverage at Length $K$ metric to measure the distribution of explanation lengths within a given behavior $B$. For a specific threshold $K$, this metric is calculated as the cumulative sum of the individual coverages of all explanations consisting of $K$ or fewer concepts, normalized by the total number of explanations across all lengths. A high score at a small $K$ indicates that the majority of explanations are short and easy to interpret.

Figure 18 plots the Relative Cumulative Individual Coverage against the Length $K$ for each evaluated behavior in the main text. Across all cases, the majority of explanations consist of $K \leq 8$ concepts, showing that our approach aligns with the range previously identified as optimal for human accuracy  Ramaswamy et al.. Among these results, LEACE CXp and Ortho AXp report the highest Cumulative Frequencies for small $K$ values, indicating they are the most concise. In contrast, Ortho CXp and SpLiCE AXp tend to produce the longest explanations in most cases.

## G.7   Signed Concept-based Explanations

The following three tables provide signed ConXps, each containing one or more pairs of concepts and their corresponding polarities as described in Definition 3.8, with the highest individual coverage among explored behaviors; note that to aid comprehension when reporting results, we list ConAXps and ConCXps separately for each erasure method but collapse by enumeration algorithm. Firstly, Table 7 reports **ResNet18-based behaviors**. Note that while several explanations achieve high individual coverages between 85% and 90%, the specific concepts vary depending on the erasure algorithm employed. For example, in model $M_2$ fine-tuned on RIVAL10, the two most frequent ConAXps produced with Ortho for behavior $B_{M_2}(\text{Car})$ are {Flight(-)} and {Taxi(+)}; while the two most frequent ConAXps found with LEACE are {Wheel(+)∧Barrier(-)} and

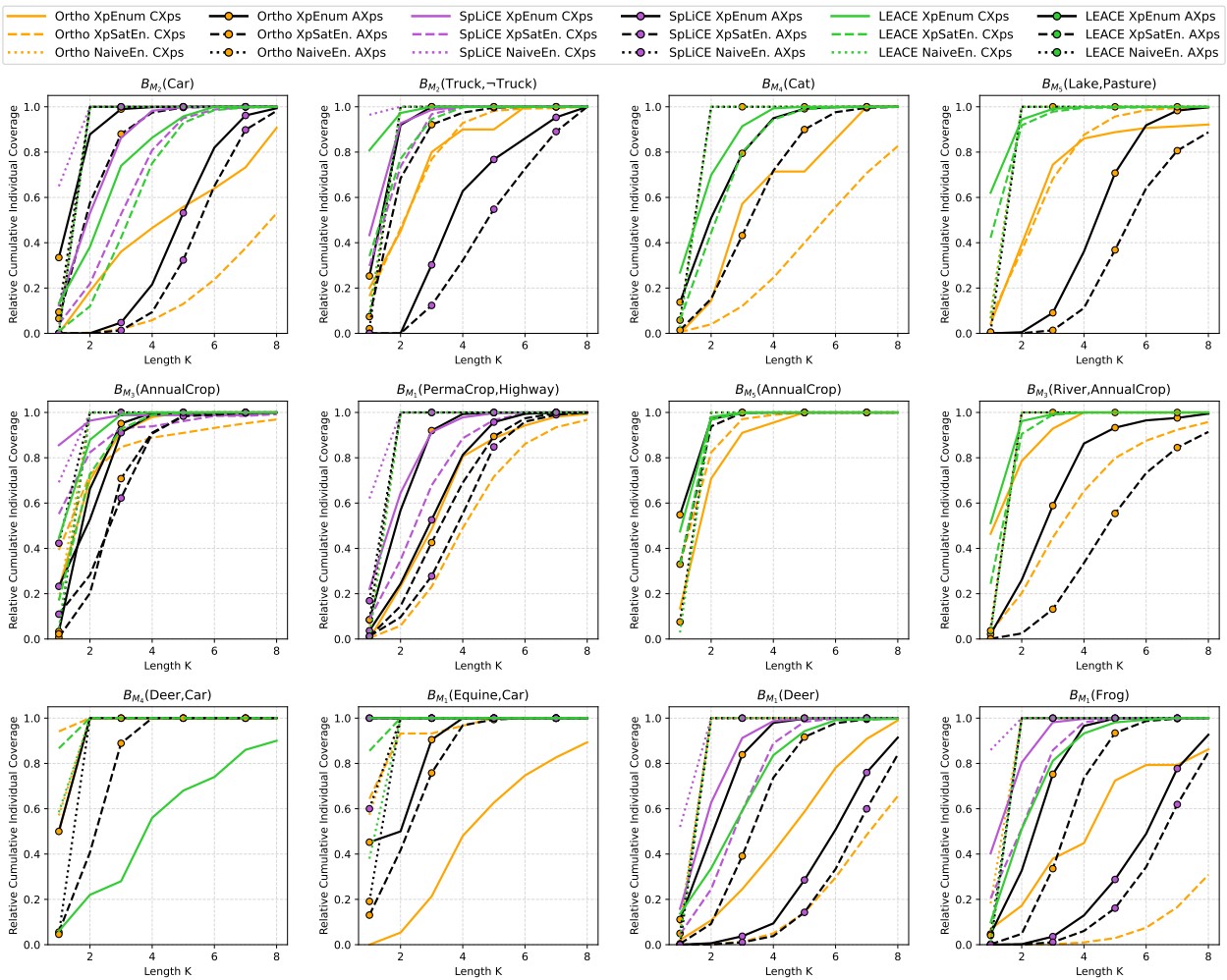

Figure 18: Relative Cumulative Frequency at Length $K$ for all model behaviors we analyze.

{Civic(+)}. In the case of incorrectly classified trucks, $B_{M_2}$(Truck,¬Truck), the most frequent ConCXps are {Taxi(+)}, {Van(+)}, and {Civic(+)} with individual coverages of 55%, 65%, and 75%, respectively.

Secondly, Table 8 shows signed ConXps for **VGG19-based behaviors** obtained via Ortho and LEACE erasure algorithms; see Section 5.2 for a discussion on why SpLiCE is not listed alongside VGG-based behaviors. Model $M_5$ fine-tuned on EuroSAT consistently reports explanations with very high individual coverages, within the [99%, 100%] range, such as {Sea(+)}, {Smoke(+)}, and {Dust(+)}. In contrast, the model $M_4$ fine-tuned on RIVAL10 struggles to do so. Its most frequent explanation accounts for only 57% of behavior $B_{M_4}$(Cat), while the runner up covers 35%, {Milk(+)} and {Agricultural(-)}, respectively. The misclassification behavior $B_{M_4}$(Deer, Car) reports four ConCXps with 100% of individual coverage: {Traffic(+)}, {Parking(+)}, {Van(+)}, and {Bus(+)}.

Lastly, Table 9 indicates that there is no clear winner on which erasure algorithm consistently produces the explanations with the highest individual coverage for **zero-shot CLIP-based behaviors**. It is interesting to note that this model shows a greater tendency to reuse the same concepts across different erasure algorithms than other models. For instance, in behavior $B_{M_1}$(Deer), the most frequent ConAXp and ConCXp are both {Hunting(+)} via Ortho and LEACE, with relative frequencies of 74% and 58%, respectively. Similarly, in the case of behavior $B_{M_1}$(Frog), the most frequent ConAXp and ConCXp are both {Reptile(+)} via Ortho and SpLiCE, with relative frequencies of 94% and 99%, respectively. Concept reuse also happens in behavior $B_{M_1}$(Equine, Car), where {Taxi(+)} and {Driver(+)} appear frequently; note that the concept

Table 7: Signed explanations with highest individual coverage for ResNet18-based behaviors.

| | Setting | | Top Signed Concept-based Explanations |
|---|---|---|---|
| $B_{M_2}$(Car) | Orth | AXp | {Flight(-)}x0.88,{Taxi(+)}x0.86,{Drive(+)∧Pet(-)}x0.85 |
| | | CXp | {Drive(+)∧Flight(-)∧Animal(-)∧Wild(-)∧Hunter(-)∧Farm(-)}x0.19 |
| | SpL | AXp | {Wagon(+)∧Continental(+)∧Golf(+)∧Civic(+)}x0.17 |
| | | CXp | {Civic(+)}x0.47,{Wagon(+)∧Civic(+)}x0.45 |
| | LEA | AXp | - |
| | | CXp | {Wheel(+)∧Barrier(-)}x0.89,{Civic(+)}x0.88 |
| $B_{M_2}$(Truck,¬Truck) | Orth | AXp | {Taxi(+)}x0.63,{Tail(-)}x0.45,{Whip(+)}x0.45,{Wagon(+)∧Pet(-)}x0.45 |
| | | CXp | {Taxi(+)}x0.55,{Whip(+)}x0.48,{Civic(+)∧Flight(-)}x0.33 |
| | SpL | AXp | {Wagon(+)∧Van(+)∧Civic(+)}x0.10,{Van(+)∧Golf(+)∧Civic(+)}x0.10 |
| | | CXp | {Van(+)}x0.65,{Civic(+)}x0.38,{Wagon(+)}x0.30,{Driver(+)}x0.23 |
| | LEA | AXp | - |
| | | CXp | {Civic(+)}x0.75,{Taxi(+)}x0.70,{Van(+)}x0.65,{Golf(+)}x0.58 |
| $B_{M_3}$(River,Crop) | Orth | AXp | {Villa(+)∧Kettle(+)}x0.30,{Villa(+)∧Shaft(+)}x0.30 |
| | | CXp | {Villa(+)}x0.38,{Villa(+)∧Peanut(+)}x0.28 |
| | LEA | AXp | - |
| | | CXp | {Solar(+)∧Fast(+)}x0.80,{Villa(+)∧Fast(+)}x0.78 |
| $B_{M_3}$(AnnualCrop) | Orth | AXp | {Plate(+)}x0.99,{Tennis(+)∧Box(+)}x0.98,{Board(-)∧Box(+)}x0.91 |
| | | CXp | {Brown(+)}x1.0,{Board(-)}x1.0,{Box(+)}x1.0,{Tennis(+)}x1.0 |
| | SpL | AXp | {Plate(+)}x0.71,{Plate(+)∧Field(+)}x0.17 |
| | | CXp | {Plate(+)}x0.96,{Plate(+)∧Field(+)}x0.06 |
| | LEA | AXp | - |
| | | CXp | {Plate(+)}x0.96,{Brown(+)}x0.95,{Box(+)}x0.91,{Train(+)}x0.91 |

ROAD appears both positively and negatively in SpLiCE and Ortho-based ConAXps, respectively. Finally, behavior $B_{M_1}$(Crop, Highway), observed in the EuroSAT dataset, also shows partial overlap of concepts; being {Solar(+)} and {Solar(+)∧Court(+)} the most frequent ConAXp and ConCXp respectively, both found via Ortho, reporting relative frequencies of 83% and 92%, respectively. It is interesting to note that this is the only behavior with LEACE × ConAXps, although their individual coverage is a negligible ≈ 2%.

## G.8 Qualitative Analysis

We explore how ConXps can assist human analysts in evaluating vision models' behaviors through two use cases. First, we find how models disambiguate concepts with multiple potential meanings. Second, we perform pixel-space transformations based on ConXps and report the model's response.

### G.8.1 Polysemantic Disambiguation

We have found that our explanations are able to identify polysemous terms and disambiguate them. In the $B_{M_1}$(Ship) behavior, while {Bay(+)} is a frequent explanation, so is the conjunction {Bay(+) ∧ Velvet(-)}, where BAY is positive and VELVET is negative. This explanation suggests that the model may initially associate BAY with its usage in equine contexts, such as the popular horse color. The requirement for VELVET to be negatively present serves to discard this animal-related meaning, effectively grounding the explanation in the maritime domain. Similarly, for $B_{M_2}$(Deer), our method found the explanation {Racks(+) ∧ Industrial(-)}. This indicates that the model recognizes the potential for RACKS to refer to industrial storage, with the negative concept ensuring the explanation remains relevant to the zoological context. In a similar fashion,

Table 8: Signed explanations with highest individual coverage for VGG19-based behaviors.

| | Setting | | Top Signed Concept-based Explanations |
|---|---|---|---|
| $B_{M_5}$ (Lake,Past.) | Orth | AXp | {Fog(+)∧Smoke(+)∧Comet(+)∧Duck(+)}x0.96,{Fog(+)∧Smoke(+)}x0.56 |
| | | CXp | {Sea(+)}x1.0,{Ocean(+)}x0.99,{Fog(+)}x0.98,{Mist(+)}x0.98,{Mosquito(+)}x0.97 |
| | LEA | AXp | - |
| | | CXp | {Smoke(+)}x1.0,{Comet(+)}x1.0,{Dust(+)}x1.0,{Ocean(+)}x1.0,{Sea(+)}x1.0 |
| $B_{M_5}$ (AnnCrop) | Orth | AXp | {Modest(+)}x0.96,{Memorial(+)}x0.90,{Jacket(+)}x0.88,{Ghost(+)}x0.86 |
| | | CXp | {Modest(+)}x0.90,{Memorial(+)}x0.90,{Jacket(+)}x0.78,{South(+)}x0.74 |
| | LEA | AXp | - |
| | | CXp | {Beach(+)}x0.92,{Kite(+)}x0.90,{Direct(+)∧Daylight(+)}x0.88,{Sky(+)}x0.86 |
| $B_{M_4}$ (Cat) | Orth | AXp | {Milk(+)}x0.57,{Steak(+)∧Smoke(+)}x0.11,{Steak(+)∧Owl(+)}x0.09,{Wild(+)}x0.09 |
| | | CXp | {Milk(+)}x0.16,{Owl(+)}x0.14,{Smoke(+)}x0.14,{Wild(+)∧Smoke(+)∧Sail(+)}x0.04 |
| | LEA | AXp | - |
| | | CXp | {Agricultural(-)}x0.35,{Moon(+)}x0.15,{Ambulance(-)}x0.15,{Wing(-)}x0.15 |
| $B_{M_4}$ (Deer,Car) | Orth | AXp | {Wolf(+)∧Pier(-)}x0.80,{Parking(+)∧Wild(+)}x0.60,{Driver(+)}0.60 |
| | | CXp | {Wild(+)}x0.80,{Lion(+)}x0.80,{Pasture(+)}x0.80,{Extinct(+)}x0.80 |
| | LEA | AXp | - |
| | | CXp | {Traffic(+)}x1.0,{Parking(+)}x1.0,{Van(+)}x1.0,{Bus(+)}x1.0,{Road(+)}x0.80 |

we observed that in $B_{M_4}$(Plane), the explanation {Flight(+) ∧ Nest(-)} distinguishes PLANE images from other flying entities like the BIRD class in the RIVAL10 dataset, successfully removing ambiguity.

> **Finding:** ConAXps and ConCXps are able to naturally disambiguate polysemous terms without explicit supervision. They highlight that the vision model also potentially internally separates different semantic meanings of polysemous concepts to enable precise classification.

### G.8.2 Pixel-space Transformations

Obtaining a ConCXp implies that erasing specific concepts from a model's internal representation will necessarily shift its prediction. To evaluate the model's response to pixel-level interventions, we conducted an experiment using a diffusion model Wu et al. (2025) to perform semantic edits directly on the input images. We tested whether erasing these concepts at the pixel level would flip the model's prediction, mimicking the effect of concept erasure. We transformed the pixels corresponding to all concepts in the ConCXps {Sea(+)∧Freight(+)}, {Racks(+)∧Wildlife(+)}, {Driver(+)∧Parking(+)}, and {Marsh(+)∧Aquarium(+)} from images of the classes SHIP, DEER, CAR, and FROG, respectively; as shown in Figure 19. Figure 20 contains additional examples for the DEER, CAR, and FROG classes and their corresponding prompt template. We also explored alternative pixel replacements for the SEA concept in SHIP images, as shown in Figure 21. Instead of replacing these pixels with white, we substitute them with textures representing DIRT or GRASS.

Despite these semantic interventions, 96% of the modified images across all behaviors are still correctly predicted as their original classes, e.g., replacing SEA and FREIGHT-related pixels in SHIP images did not change the model's output. To analyze why, we provided the modified images through the vision encoders and measured the cosine similarity between the resulting embeddings and the concept vectors, as defined in Section 2.2.1 and Appendix D. We found that concepts removed at the pixel level remain positively present at the embedding level. This contrasts with erasure algorithms applied directly to the embeddings, which reduce concept strength to zero. The model's correct classification is justified because the concepts required for the prediction remain present in the embedding despite the pixel-level changes. These results support the finding that vision encoders infer and encode concepts from contextual cues, even when the specific pixels representing those concepts are replaced.

Table 9: Signed explanations with highest individual coverage for zero-shot CLIP-based behaviors.

| | Setting | | Top Signed Concept-based Explanations |
|---|---|---|---|
| $B_{M_1}$ (Deer) | Orth | AXp | {Hunting(+)}x0.74,{Wildlife(+)}x0.64,{Rack(+)∧Forest(+)}x0.62 |
| | | CXp | {Wildlife(+)∧Hunting(+)∧Rack(+)∧Forest(+)}x0.24 |
| | SpL | AXp | {Horn(+)∧Hunter(+)∧Rack(+)∧Reserve(+)∧Goat(+)}x0.06 |
| | | CXp | {Goat(+)}x0.41,{Hunting(+)}x0.34,{Rack(+)}x0.31,{Calf(+)}x0.25 |
| | LEA | AXp | - |
| | | CXp | {Hunting(+)}x0.58,{Hunter(+)}x0.48,{Rack(+)∧Industrial(-)}x0.39 |
| $B_{M_1}$ (Equine,Car) | Orth | AXp | {Pasture(+)∧Wild(-)}x0.67,{Wild(+)∧Road(-)}x0.67 |
| | | CXp | {Wild(+)}x0.67,{Taxi(+)}x0.67,{Forest(+)}x0.44,{Lion(+)}x0.44,{Grass(+)}x0.33 |
| | SpL | AXp | {Taxi(+)}x0.44,{Road(+)}x0.11,{Driver(+)}x0.11,{Cart(+)∧Road(-)}x0.11 |
| | | CXp | {Taxi(+)}x1.0,{Driver(+)}x0.78,{Traffic(+)}x0.78,{Trunk(+)}x0.78 |
| | LEA | AXp | - |
| | | CXp | {Taxi(+)}x1.0,{Driver(+)}x1.0,{Parking(+)}x0.78,{Traffic(+)}x0.78 |
| $B_{M_1}$ (Frog) | Orth | AXp | {Reptile(+)}x0.94,{Snail(+)∧Wheat(-)}x0.57,{Insect(+)∧Wheat(-)}x0.55 |
| | | CXp | {Reptile(+)∧Insect(+)∧Snail(+)∧Pigeon(-)∧Beef(-)∧Wheat(-)}x0.10 |
| | SpL | AXp | {Reptile(+)∧Marsh(+)∧Snail(+)∧Cricket(+)∧Aquarium(+)}x0.02 |
| | | CXp | {Reptile(+)}x0.99,{Cricket(+)}x0.85,{Snail(+)}x0.50,{Aquarium(+)}x0.43 |
| | LEA | AXp | - |
| | | CXp | {Insect(+)∧Monarch(-)}x0.78,{Cricket(+)∧Feather(-)}x0.69,{Reptile(+)}x0.63 |
| $B_{M_1}$ (Crop,Highway) | Orth | AXp | {Solar(+)}x0.83,{Court(+)∧Navy(+)}x0.72,{Court(+)∧Frog(-)}x0.70 |
| | | CXp | {Solar(+)∧Court(+)}x0.92,{Solar(+)}x0.51,{Flag(+)∧Court(+)}x0.32 |
| | SpL | AXp | {Solar(+)∧Navy(+)}x0.26,{Solar(+)}x0.19,{Solar(+)∧Court(+)}x0.18 |
| | | CXp | {Solar(+)}x0.40,{Solar(+)∧Court(+)}x0.39{Solar(+)∧Navy(+)}x0.27 |
| | LEA | AXp | {Solar(+)∧Display(+)}x0.02,{Solar(+)∧Court(+)}x0.02 |
| | | CXp | {Court(+)∧Bus(+)}x0.56,{Level(+)∧Court(+)}x0.52,{Solar(+)∧Spray(+)}x0.42 |

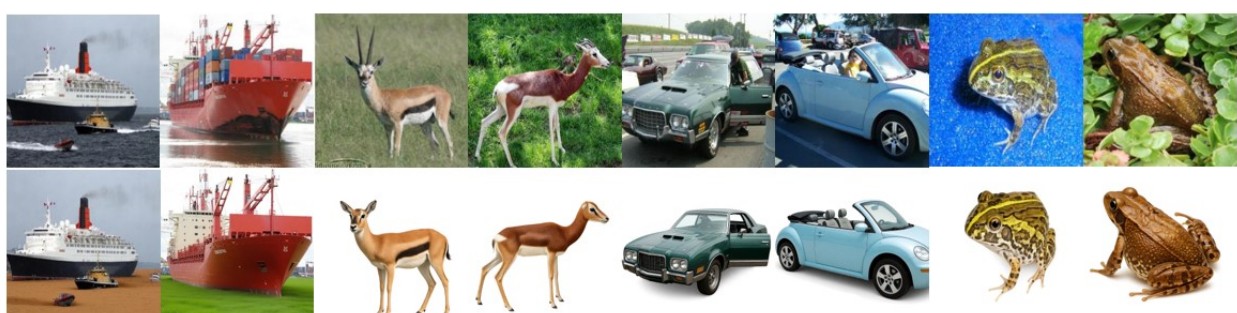

Figure 19: Pixel-space transformations performed using a diffusion model to remove concepts from images based on ConCXps extracted from them. We evaluated vision models' response to these targeted interventions.

**Finding:** Vision encoders potentially infer and encode concepts from contextual cues despite the erasure of the corresponding pixels in the image.

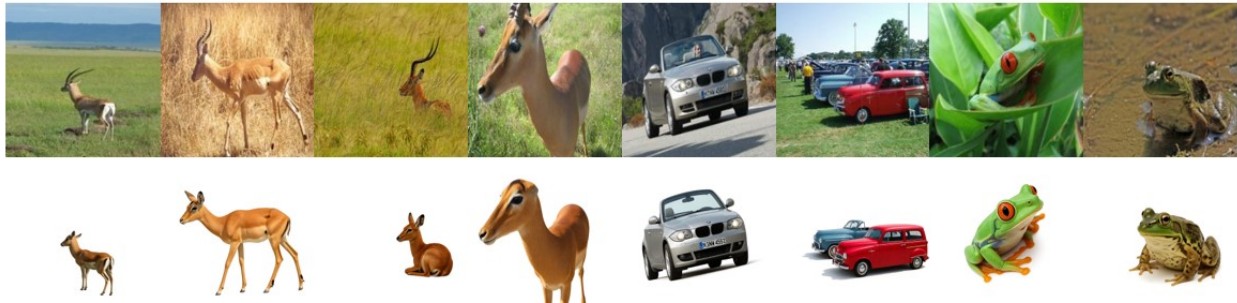

Figure 20: Supplementary examples for pixel-space transformations. Prompt used: 'Remove the {RACKS| DRIVER|MARSH}. Remove {WILDLIFE|PARKING|AQUARIUM} background. Replace with a white background.'

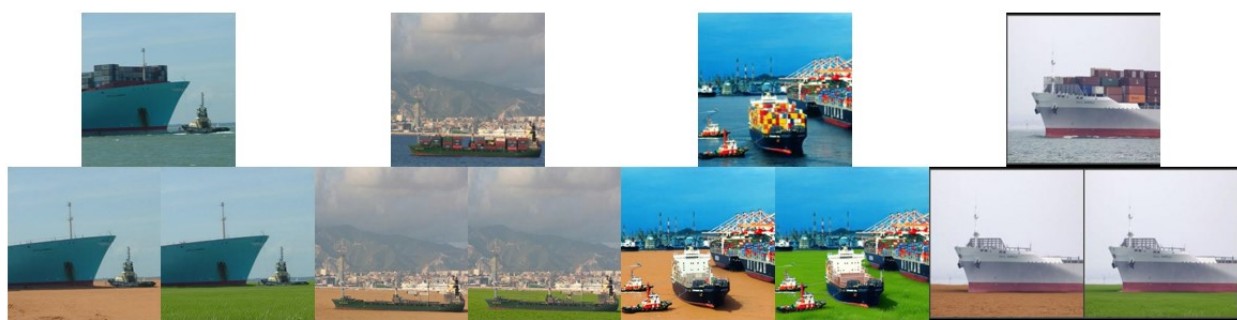

Figure 21: Supplementary examples for pixel-space transformations. Prompt used: 'Remove the {SEA} under the {SHIP}. Replace with {DIRT|GRASS}. Remove {FREIGHT} on the {SHIP}.'

# H  *LLM-as-a-judge* Implementation

As described in Section 5.2.4, we categorize each concept in our base vocabulary as relevant or irrelevant in our plausibility evaluation, using Gemini 3 Pro Google DeepMind (2026), following the *LLM-as-a-judge* strategy Li et al. (2025). This appendix provides the exact prompts used to generate these sets. The instructions are specific to the type of behavior under consideration: correct classifications, where ground truth equals the prediction, and misclassifications, where the model confuses the ground truth with a different class.

### H.1 Correct Classification Behavior Instructions

> **Role:** You are acting as an expert in Computer Vision and Explainable Artificial Intelligence (XAI). Your task is to evaluate the semantic relevance of concepts within a predefined vocabulary relative to a vision model's classification behavior.
> **Context:** A vision model has correctly classified an image.
> - Ground Truth Label: {class_a}
> - Model Prediction: {class_a}
> **Task:** You will be provided with a vocabulary of {N} concepts. Classify each concept as either RELEVANT or IRRELEVANT for the prediction of {class_a}.
> **Evaluation Criteria:**
> - RELEVANT: A concept is relevant if its presence provides useful information about the class. Examples: 'wheels' is relevant for 'car'; 'horns' is relevant for 'deer'; 'wet' is relevant for 'ship'.
> - IRRELEVANT: Concepts that are semantically unrelated; these concepts provide no information about the class. Examples: 'rectangular' is irrelevant for 'cat'; 'text' is irrelevant for 'bird'.
> **Vocabulary:** [comma-separated list of N words]
> **Output:** RELEVANT: [comma-separated list], IRRELEVANT: [comma-separated list].

### H.2 Misclassification Behavior Instructions

> **Role:** You are acting as an expert in Computer Vision and Explainable Artificial Intelligence (XAI). Your task is to evaluate the semantic relevance of concepts within a predefined vocabulary relative to a vision model's incorrect classification behavior.
> **Context:** A vision model has misclassified an image.
> - Ground Truth Label: {class_a}
> - Model Prediction: {class_b}
> **Task:** You will be provided with a vocabulary of {N} concepts. Classify each concept as either RELEVANT or IRRELEVANT in explaining the confusion between {class_a} and {class_b}.
> **Evaluation Criteria:**
> - RELEVANT: Concepts semantically associated with either {class_a} or {class_b}. Example: 'wings' is relevant when mistaking a 'frog' (ground truth) for a 'bird' (prediction).
> - IRRELEVANT: Concepts semantically unrelated to both {class_a} and {class_b}. Example: 'metallic' is irrelevant when mistaking a 'cat' (ground truth) for a 'dog' (prediction).
> **Vocabulary:** [comma-separated list of N words]
> **Output:** RELEVANT: [comma-separated list], IRRELEVANT: [comma-separated list].

