# OpenReview forum: "Concept-Based Abductive and Contrastive Explanations for Behaviors of Vision Models"
_TMLR — Under review for TMLR_

### Review · Reviewer_BVdN · 2026-07-07

**Summary Of Contributions:**

**Summary**

The authors present a method for generating causal, context-based explanations for image classifiers. The main contribution is the integration of formal counterfactual analysis to derive abductive and contrastive concept-based explanations. While similar analyses are typically performed in the low-level input feature space (e.g., pixel perturbations), the proposed approach transposes the analysis to a latent concept representation, enabling explanations at a more semantically meaningful level.

**Strengths**

- Provides abductive and contrastive concept-based explanations.
- Introduces a novel application of formal counterfactual analysis in a latent concept space, moving beyond traditional pixel-level perturbation methods toward more semantically meaningful explanations.
- Includes a comprehensive empirical evaluation across multiple classifiers, concept-erasure techniques, search strategies, and evaluation metrics.

**Weaknesses**

- The causal interpretation relies heavily on the CLIP semantic space used as an intermediary representation, making it unclear whether the identified explanations reflect causal mechanisms of the target classifier or semantic relationships encoded in CLIP.
- Concepts that are semantically related to a target class may receive high causal attribution due to their proximity in the CLIP embedding space, even when they do not correspond to independent causal factors used by the classifier.

**Audience:**

Yes

**Audience Explanation:**

The attempt to uncover causal relationships underlying model decisions at the concept level is both timely and compelling. The paper addresses an important challenge in explainable AI by combining concept-based explanations with formal counterfactual reasoning. While the current evaluation does not fully substantiate the causal interpretation of the identified concepts, the proposed direction is promising and highlights several important research questions. This work has the potential to encourage further research toward more rigorous and reliable frameworks for concept-level causal explanations.

**Broader Impact Concerns:**

N/A. The paper does not raise any significant ethical concerns beyond those typically associated with research on explainable and interpretable AI systems.

**Claims And Evidence:**

No

**Claims Explanation:**

The proposed framework relies on a mapping between the classifier's latent representation and the CLIP embedding space. Consequently, the validity of the resulting explanations depends strongly on the quality and faithfulness of this alignment. While the experiments demonstrate that meaningful concept-based explanations can be generated, they do not sufficiently establish that the identified concepts correspond to causal mechanisms of the target classifier itself.

As a result, the explanations may primarily reflect the semantic structure and biases encoded in CLIP rather than causal mechanisms intrinsic to the target classifier. For example, although the classifier may be trained to recognize classes such as *deer*, *hunting* could emerge as an influential explanatory concept simply because deer images are strongly associated with hunting-related content in the CLIP embedding space, rather than because hunting causally contributes to the prediction.

More generally, concepts that are semantically close to a target class may receive high causal attribution despite not representing independent causal factors used by the classifier. For example, a concept such as *vessel* may emerge as highly influential for the class *ship* largely because both concepts occupy nearby regions of the semantic space. In such cases, the framework may recover semantic relationships encoded in CLIP rather than causal explanations of the classifier's decision-making process.

The authors partially address this issue by removing concepts with cosine similarity greater than 0.9. However, many strongly related concepts are likely to remain below this threshold while still exhibiting substantial semantic overlap with the target classes. The paper does not provide sufficient evidence that the identified explanatory concepts are not primarily driven by such semantic associations.

Overall, the empirical results demonstrate that the framework can generate meaningful concept-based explanations and identify semantically relevant concepts. However, they do not sufficiently support the stronger claim that the discovered concepts correspond to causal mechanisms of the target classifier rather than semantic associations inherited from the CLIP representation.

**Requested Changes:**

**Critical Revisions**

### 1. Provide evidence that the discovered explanations are not primarily driven by the CLIP semantic representation.

The proposed framework relies on CLIP as an intermediary semantic space between the classifier representation and the concept vocabulary. Consequently, it remains unclear to what extent the identified explanatory concepts reflect the behavior of the target classifier rather than properties of the CLIP representation itself.

To strengthen the causal interpretation, the authors should compare the discovered explanatory concepts with those obtained using established concept-based explanation methods that rely on external concept vocabularies or probe datasets (e.g., TCAV-style approaches or concept-based probing methods). While exact agreement would not be expected, a substantial overlap would provide evidence that the explanations are not solely a consequence of the semantic relationships encoded in CLIP.

### 2. Quantify and analyze concept–class correlations.

The causal interpretation of the identified concepts depends critically on the degree of semantic overlap between concepts and target classes. Concepts that are strongly related to a class label may receive high causal attribution simply due to their proximity in the semantic space rather than because they correspond to independent causal factors used by the classifier.

The authors should therefore quantify and report concept–class similarity across the vocabulary (e.g., cosine similarity between concept embeddings and class-label embeddings) and analyze the relationship between these similarities and the resulting causal attribution scores. A strong correlation would suggest that the method primarily recovers semantic associations already encoded in the representation space rather than identifying independent causal mechanisms underlying the classifier's decisions.

---

### Review · Reviewer_kHKu · 2026-07-07

**Summary Of Contributions:**

The paper proposes an approach that captures the elements appearing in an image as vision/language-based concepts and, building on these concepts, computes abductive and contrastive explanations. In particular, the paper formalizes the targeted forms of explanation and then proposes, in order to handle concepts in images, an erasure algorithm that removes the components corresponding to concepts from image embeddings, together with algorithms for efficiently enumerating the explanations.

**Additional Comments:**

While reading the paper, I had the following questions. These are intended as comments rather than requests for revision, but I believe the paper would be even more helpful to readers if these points were additionally discussed.

(1) Section 2.2.1 discusses the linear maps between the vision model and CLIP embedding spaces ("Mapping from Vision Models to CLIP and Back"), and these maps are validated through the sanity checks in Appendix D.1. Could there be cases in which this mapping does not work well? If so, how should such cases be handled?

(2) Related to (1): rather than training a vision encoder of one's own, would it be meaningful to use the CLIP image encoder as is, or to fine-tune it, for example by adding new layers on top of the CLIP encoder and training only those layers?

(3) In the formulation, each image embedding $z$ is viewed as being represented by a subset of the concepts ${c_1, \dots, c_n}$. Do these concepts need to be "disjoint"? That is, is it acceptable for the notion represented by one concept to overlap with the notion represented by another concept? And if such overlaps are large or frequent, would this have any adverse effects on the explanation pipeline?

(4) The experiments evaluate combinations of the different erasure algorithms and enumeration algorithms. For the enumeration algorithms, it is easy to imagine that the choice affects the efficiency of the search. On the other hand, what accounts for the differences in the results across the erasure algorithms, and which erasure algorithm should one use in what cases? I believe it would be helpful if the paper commented on these points.

(5) Figure 9 uses the term "valid concepts" (in the definition of the Validity Ratio given in the caption). What does this term refer to?

Minor points:

(6) In some places the citation style appears to be incorrect; citations should be properly placed in parentheses where appropriate, for example by using the \citep command.

(7) Page 4, line 2: "in their representation spaces These concept ..." a period appears to be missing after "spaces".

**Audience:**

Yes

**Audience Explanation:**

Model behavior explanation is of certain interest broadly in machine learning. The paper presents an interesting, technically solid approach for conecpt-based, visual model explanation.

**Broader Impact Concerns:**

As noted above, the limitation of the proposed method could have been elaborated more.

**Claims And Evidence:**

Yes

**Claims Explanation:**

The proposed method's validity and runtime efficiency are supported nicely by the empirical results.

**Requested Changes:**

The motivation, the formalization, the proposed methods, and the experiments are all treated in technical depth and are of a high standard. I did not find any major issues, and I believe the paper can be accepted after minor corrections. See the Additional Comments section below for points that may appreciate more discussion in the paper. Also, the  limitation of the proposed method could have been discussed in more details, perhaps in the conclusion section.

---

### Review · Reviewer_4tZj · 2026-07-13

**Summary Of Contributions:**

This paper aims to explain the prediction of an image classifier by stating which global abstract concepts are sufficient to maintain the prediction (abductive explanation), and which are sufficient to change in order to change the prediction (contrastive explanation).
The key contribution of this work is to achieve this by combining for the first time methods from concept-based explainability with formal diagnosis of knowledge bases.
This mostly relies on existing techniques for building concept bases, extracting and erasing concepts from latent spaces, notions of abductive and contrastive explanations, and formal techniques to search for them.
Two novel additions are
1. a proposal for simplifying the diagnosis problem using (unfortunately rather strong and insufficiently empirically evidenced) assumptions;
2. a concept erasure technique that incorporates orthogonality constraints (similar to concept whitening); and
3. a method for post-hoc pruning obtained sets of abductive and contrastive explanations to high-coverage subsets of size suitable for presentation to human explainees.

**Additional Comments:**

The proposed method is well motivated, and the paper is generally well-written and easy to follow.

Some minor findings / nice-to-have:
- The choice of naming for the explanation types, ConAXp and ConCXp, is visually so similar that they are easy to confuse. It might be better to use shorter and more different abbreviations, like simply CX and AX.
- It would be instructive to see results also on more concurrent smaller vision architectures, like ConvNeXt or Vision Transformers, as well as more datasets like the discussed medical use-case.
- The method and evaluation directly aims at easy-to-read rules that are useful for explainees. While I agree that rule complexity as well as plausibility tend to be good proxy measures for usefulness, a user study that explicitly tests this on real users would be way more convincing.
- Z is often used for denoting signed integers, so it would avoid confusions to use a different symbol to denote the representation space.
- In footnote 2 it is stated that projection of the embeddings onto the concept vectors is an alternative method to calculating cosine similarity. However, these are (approximately) the same for normalized concept vectors and CLIP embeddings or embeddings subject to batch normalization.
- In Definition 3.3 (weak contrastive explanation) it says "if their presence or absence is changed". In the remainder of the definition it is not clear where concepts are turned present and where turned absent. It might make more sense to here not refer to concepts, but simply literals.
- Def. 3.5 (Monotonicity of Head) formulates as intuition: "if adding a concept to a representation". This is not intuitively clear (what is "adding" here?). I would recommend to be more precise, e.g., "if turning a concept from absent to present in the representation".
- In the intro to Sec. 4, RQ4 mentions relevance and irrelevance. These terms have insufficiently been introduced to this point.
- In several places, the wrong citation commands seem to have been used, since non-inline-style citations appear across the paper.
- Tab. 5: Here it compares concept importances in images with concept c1 and no concept c2. By rationale of the paper introduced before, the strength of absent concepts should be lower than that of present ones. This is not the case here. Why?
- Tab. 5: Please add standard deviation.

**Audience:**

Yes

**Audience Explanation:**

This combination of classical diagnosis with concept-based explanations is very interesting. It will not only spark interest into the merging of these directions, but also shows initial results and concrete implementation ideas.

This is why I would like to highly encourage the authors to clarify the remaining unclarities regarding the monotonicity assumption. Even if this is not widely applicable in practice, if respectively discussed, it may still be a valuable contribution to push the field.

**Broader Impact Concerns:**

The monotonicity assumption will hardly be applicable in practice, see critique above.

**Claims And Evidence:**

No

**Claims Explanation:**

While the computational effort justifies this, the paper solely relies on experiments on 3 network architectures, two of which are particularly small, and one being a not further specified CLIP variant.

However, my most severe doubt is the key contribution of a problem simplification that strictly relies on a monotonicity assumption. This monotonicity assumption essentially states that if only a subset of concepts in an input is present and already yield the same output $k$ as the original input, then "adding" (i.e., turning on presence) of any other concept will not change the output.
This explicitly ignores feature interactions of the type $(a\wedge \neg b)\to k$. A simple but critical real-world example would be a situation where a car has to stop (k) at a crossing with street light (a) which signals green (b) because of an ambulance (c). Then $(a\wedge b\wedge c)\to k$, and $(a\wedge \neg b\wedge\neg c)\to k$, but $(a\wedge b\wedge \neg c)\not\to k$.
From a different perspective, to what I understand the fundamental issue here is a discrepancy between the original diagnosis approach and the erasure approach presented here: The erasure approach assumes that the boolean value of every concept which does not appear in the explanation is false (i.e., absent / erased; encoded by $b_j=0$) and that in particular "negative concepts are irrelevant for the behavior". This makes it comparatively simple to verify whether entailment properties hold. Whereas in diagnosis, one would not assume any information about those non-appearing concepts, but instead demand that entailment holds for any values of those unmentioned concepts. I acknowledge that the monotonicity assumption is a correct and neat trick to drastically reduce search. However, it is unsuprisingly not applicable in practice, as both shown in the experimental validation in Sec. A.1, as well as there acknowledged by the authors ("monotonicity is not strictly satisfied in either direction"). This also goes in line with the finding that "a significant proportion of explanations are only partially plausible".
Sacrificing the capability to capture feature interactions disregards the key advantage over way cheaper classical perturbation-based feature importance methods. These implications are currently not discussed nor argued.

Also, while the proofs in Sec. A are correct, and the key insights from the crucial empirical validation are formulated correctly, the extent of the experiments and the detailed results analysis here is questionable. To what I understood, a test on only five concepts was conducted. This also was not exhaustive, but relied on random perturbations in concept space, which does not take into account the similarity in image space and could therefore overemphasize unrealistic perturbations. Where this small study does not justify any confident claims in my opinion, a failure rate of mostly more than 10%, partly even more than 40% for abduction, cannot be called "the great majority [...] supporting [...] the simplified-to-universal lift our algorithms rely on".

**Requested Changes:**

## Clarifications and Justifications of the Approach

1. **Monotonicity assumption:** Please provide more diverse empirical evidence (more than 5 concepts) that this is applicable in practice. Please also discuss its limitations and where in practice issues are to be expected (best also proven by respective empirical study).

2. **Concept erasure:** How are concept dependencies (e.g., hierarchies) taken into account for erasure? E.g., Fig. 1 nicely illustrates that some superconcepts get erased in order to erase the subconcepts, such as erasing "pet" to not get "cat" as output, or erasing "wolf" to not get "dog" as output. This should be clarified in the main paper.

3. **Discretized explanations:** Def. 3.8 suggests to use signed explanations, i.e., to discretize per-concept influence values into positive, negative, and neutral influence. While this is known practice from techniques like TCAV for reducing to a stable single value, this binning is instead later used to calculate a histogram. It is unclear why one should not directly use the continuous concept influence values for calculating the histogram. Furthermore, it looks like the value 0 can only be taken in case of exact 0 influence. This sounds like an instable solution, where typically a bin is positioned around 0 with two cutoff thresholds. It should be clarified in the main paper how this is implemented.

4. **Baseline depth:** The search depth of the baseline is chosen particularly small with explanations of length only 2. It should be discussed, in how far this infringes comparability against the alternative approaches, since very different explanations might be received.

5. **Explanation similarity in metrics:** The generalizability metric introduced in Def. 5.1 only considers exact matches of explanations. I am wondering whether lengthy explanations that only differ slightly, e.g., only in one concept, should not also be considered partly duplicates.

6. **Finding on parsimonous explanations:** in Sec. 5.2.3 it says "parsimonious explanations are more likely to capture the broad, systematic patterns of a model's behavior". This contradicts the intuition that increasing the amount of constraints that are part of an explanation should also reduce its coverage and therefore its capturing of broad patterns. Also, it contradicts the findings box saying the opposite "Short ConXps consistently report higher individual coverage".

## Structure
It would be easier to follow if the general desirables for explanations would get their own subsection in the approach section. Currently they are interweaved with the experimental results.

## Related Work
The related work section is way too short to properly position both the provided method as well as the choices of its ingredients amongst the state-of-the-art.
Most of the related work consists of a single existence statement with just a long list of references, failing to (1) make clear what the respective covered directions are that make such a long list of references necessary, and (2) what they have in common and how they concretely differ from the proposed approach. E.g., the statement "Closely related are methods performing formal reasoning over ..." is lacking any argument in terms of how they are related.
Concretely, I am missing:
1. A broader positioning within the field of explainable AI, in particular in relation to
	a. post-hoc local feature importance techniques,
	b. ante-hoc techniques, in particular intervention in concept bottleneck models as first introduced by Koh et al. http://proceedings.mlr.press/v119/koh20a.html
2. Related work on diagnosis algorithms from which the choice of the employed ones was made;
3. Related work on concept erasure, and why the two methods used in the paper were chosen from amongst that literature;
4. Works discussing polysemanticity.
5. Works discussing the linear representation hypothesis.

Apart from that, some statements and definitions in the preliminaries require better reference to their origin, also to make more clear where the authors add novelty and which parts leverage existing literature. Concretely:
- Please make clear where the definitions of contrastive and abductive explanations, and for concept erasure are taken from.
- Please provide references for the duality trick for abductive and contrastive examples and shortly explain it for self-containment.

## Experimental Settings
While the computational effort justifies this, the paper solely relies on 3 models and essentially two datasets, as well as just a single model for the concept alignment.

## Other
1. In the introduction of Sec. 2 it is claimed that "the techniques [...] are generic and applicable to models designed for other tasks."
  This requires better explanation: It is not clear to me, how the techniques should transfer to, e.g., regression or object detection. To what I know, the used techniques for finding abductive and contrastive examples require binary feature encodings both of. inputs as well as of outputs.
2. For self-containment, the formula for cosine similarity should be added to Def 2.5, as well as a short note on the value range.
3. In Sec. 3 before Def. 3.1 it is said "the feature space of $M^{head}$ is defined by $B=\prod_i\{0,1\}$". This I would interpret as (1) each concept being represented by only a single neuron, which strictly contradicts polysemanticity; and (2) that it is strictly assumed that the chosen set of concepts is complete. (1) is not the case, if I understood the appendix on concept extraction correctly, where concept vectors instead of unit directions are used. Thus, this should be clarified. And (2) is only approximately true due to the simplifications that need to be made in order to limit the amount of concepts, and is not needed for creating abductive or contrastive explanations.
4. Please specify which CLIP version (particularly, which backbone, which implementation) was used.
5. In 5.2 it says four behaviors are "in the main body of the paper", whereas the respective qualitative results are referencing to the appendix.
6. A.1 claims that "[results with the highest individual coverage] represent the most relevant explanations for human analysts". This should be more clearly argued, since complete lines of research aim to help humans identify specialized cases that indicate labeling issues or corner cases.
7. Sec. C.1, p. 25 last equation: the used symbol $e$ was not introduced.